# Learning Self-Correction in Vision–Language Models via Rollout Augmentation

Yi Ding [1]   Ziliang Qiu [2]   Bolian Li [1]   Ruqi Zhang [1]

**Project Page:** https://dripnowhy.github.io/Octopus/

## Abstract

Self-correction is essential for solving complex reasoning problems in vision–language models (VLMs). However, existing reinforcement learning (RL) methods struggle to learn it, as effective self-correction behaviors emerge only rarely, making learning signals extremely sparse. To address this challenge, we propose correction-specific rollouts (**Octopus**), an RL rollout augmentation framework that synthesizes dense self-correction examples by recombining existing rollouts. This augmentation simultaneously improves sample efficiency due to rollout reuse and stabilizes RL optimization through balanced supervision. Furthermore, we introduce a response-masking strategy that decouples self-correction from direct reasoning, avoiding signal conflicts and enabling both behaviors to be learned effectively. Building on this, we introduce `Octopus-8B`, a reasoning VLM with controllable self-correction capability. Across 7 benchmarks, it achieves SoTA performance among open-source VLMs, outperforming the best RLVR baseline by 1.0 score while requiring only $0.72\times$ training time per step.

## 1. Introduction

Vision–language models (VLMs) with reasoning capabilities (Jaech et al., 2024; Comanici et al., 2025; Anthropic, 2024) have achieved impressive performance on complex image–text tasks, including real-world perception (Zhang et al., 2024c), diagram interpretation (Masry et al., 2022), and mathematical reasoning (Lu et al., 2023). Alongside reasoning improvements, reinforcement learning (RL) has been observed to induce emergent behaviors such as self-correction over previous reasoning steps (Wang et al., 2025a;

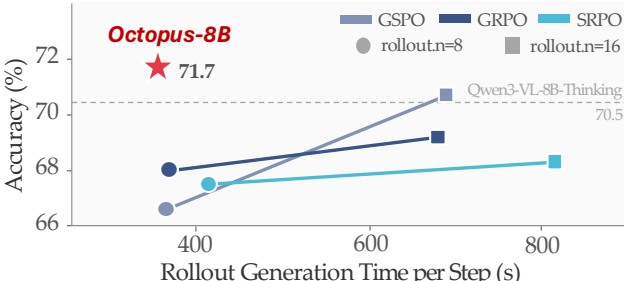

*Figure 1.* Comparison of accuracy and training efficiency across different RL methods initialized on Qwen3-8B-VL-Instruct. Octopus achieves the best average accuracy across seven benchmarks while requiring substantially less rollout time.

Jian et al., 2025), often referred to as an "aha moment" (Guo et al., 2025). These behaviors resemble how humans tackle challenging problems, suggesting that self-correction is an important capability for strong and robust reasoning.

However, current RL paradigms do not explicitly teach self-correction. Rewards are provided only at the outcome level, providing no direct signal for learning how to recover from errors. As a result, self-correction behavior arises only implicitly, is difficult to control, and cannot be reliably triggered at inference time (Ding & Zhang, 2025). This raises a central question: *how can self-correction be learned as a controllable behavior in VLMs using RL?*

Prior attempts (Wan et al., 2025) have explored encouraging self-reflection by prompting and rewarding reflective behavior during RL. While such approaches can amplify reflective tendencies, they still rely on sparse and uncontrolled emergence. Effective self-correction examples remain rare throughout training.

To address this challenge, we make a key observation: although VLMs rarely generate effective self-correction examples on their own, the necessary learning signals already exist in standard RL rollouts. For a given input, correct and incorrect self-generated reasoning trajectories often *coexist*, and their contrast naturally reveals how reasoning errors can be corrected. By pairing such trajectories, effective self-correction samples can be synthesized explicitly without additional computational overheads.

[1]Department of Computer Science, Purdue University, West Lafayette, USA [2]School of Information Sciences, University of Illinois Urbana-Champaign, Champaign, USA.

*Proceedings of the $43^{rd}$ International Conference on Machine Learning*, Seoul, South Korea. PMLR 306, 2026. Copyright 2026 by the author(s).

Based on this insight, we introduce **c**orre**ctio**n-s**p**ecific rollo**uts** (Octopus), a rollout augmentation framework for learning self-correction in RL. Octopus reuses and recombines rollouts, which not only provides dense, explicit self-correction examples, but also (i) combinatorially increases training samples (creating $n^2$ from $n$ rollouts) and (ii) balances positive and negative examples, leading to more stable policy updates. To achieve both strong self-correction and direct reasoning, we further propose a response-masking strategy that decouples their training signals and avoids optimization conflicts. Our main contributions are summarized as follows:

- We identify a key challenge in teaching self-correction via RL: effective self-correction examples are extremely sparse. We show that this challenge can be addressed by exploiting contrastive signals already present in standard RL rollouts through pairing correct and incorrect reasoning trajectories.

- We introduce Octopus, a rollout augmentation framework that constructs dense, explicit self-correction examples in RL. Octopus simultaneously improves sample efficiency via rollout reuse and stabilizes optimization by balancing positive and negative examples.

- We propose a response-masking optimization strategy that avoids conflicting training signals between self-correction and direct reasoning, enabling the model to effectively learn both capabilities.

- Our `Octopus-8B` model achieves SoTA performance among models of comparable size across 7 benchmarks. It outperforms the base model Qwen3-VL-8B-Instruct by 9.5 average accuracy points, exceeds the official reasoning-enhanced RL version Qwen3-VL-8B-Thinking by 1.2 points, and surpasses the strongest RLVR baseline, GSPO, by 1.0 average accuracy point while requiring only $0.72\times$ training time per step.

## 2. Preliminaries

### 2.1. Reinforcement Learning with Verifiable Rewards

Reinforcement Learning with Verifiable Rewards (RLVR) trains language models on tasks whose outcomes can be easily verified, such as math problems and question answering (Lambert et al., 2024). Recent studies (Yang et al., 2025; Chen et al., 2025) have shown that RLVR effectively enhances the reasoning capabilities of language models.

Group Relative Policy Optimization (GRPO) (Shao et al., 2024) is a commonly used algorithm for RLVR. It estimates advantages over a set of responses $\{o_1, \ldots, o_n\}$ generated by the policy $\pi_\theta$. The GRPO objective is defined as:

$$\mathcal{J}_{\text{GRPO}}(\theta) = \mathbb{E}_{\{o_i\}_{i=1}^G}\left[\frac{1}{G}\sum_{i=1}^G \frac{1}{|o_i|}\sum_{t=1}^{|o_i|}\min\left(w_{i,t}(\theta)\hat{A}_{i,t},\right.\right.$$
$$\left.\left.\text{clip}\left(w_{i,t}(\theta), 1-\epsilon, 1+\epsilon\right)\hat{A}_{i,t}\right)\right], \quad (1)$$

where $\hat{A}_{i,t} = \frac{r_i - \text{mean}(\{r_i\}_{i=1}^G)}{\text{std}(\{r_i\}_{i=1}^G)}$ denotes the advantage estimated from the normalized rule-based reward within the rollout group, and $w_{i,t}$ is the importance sampling ratio, computed as $\frac{\pi_\theta(o_{i,t}|x,o_{i,<t})}{\pi_{\text{old}}(o_{i,t}|x,o_{i,<t})}$. The clipping parameter $\epsilon$ prevents overly aggressive policy updates. However, scaling the number of rollouts introduces an *off-policy* effect, as rollouts must be partitioned into multiple mini-batches to compute Eq. (1). In this case, token-level importance weighting can introduce high-variance noise into the gradient estimates. Group Sequence Policy Optimization (GSPO) (Zheng et al., 2025a) mitigates this issue by applying a sequence-level importance weight $s_i(\theta) = \frac{\pi_\theta(o_i|x)}{\pi_{\text{old}}(o_i|x)}$, which prevents training collapse caused by a small number of overly off-policy tokens in long reasoning trajectories, thereby stabilizing RL training. The GSPO objective is as follows:

$$\mathcal{J}_{\text{GSPO}}(\theta) = \mathbb{E}_{\{o_i\}_{i=1}^G}\left[\frac{1}{G}\sum_{i=1}^G \min\left(s_i(\theta)\hat{A}_i,\right.\right.$$
$$\left.\left.\text{clip}\left(s_i(\theta), 1-\epsilon, 1+\epsilon\right)\hat{A}_i\right)\right]. \quad (2)$$

### 2.2. Definition of Self-Correction

Our goal is to improve the reasoning performance of VLMs by strengthening their self-correction ability. Motivated by the observation that reasoning models may spontaneously generate reflective tokens during generation, we focus on *single-pass self-correction*, where the model revises its reasoning within a single response: $(o_1 \oplus \texttt{<sc>} \oplus o_2) \sim \pi(\cdot \mid x)$, where $o_1$ and $o_2$ denote the responses before and after self-correction, and $\texttt{<sc>}$ is a special token that marks the onset of corrective behavior, e.g., an "aha moment" token (Wang et al., 2025a). Unlike multi-pass self-correction or tool-based approaches that rely on additional prompting or external feedback, this formulation treats self-correction as an intrinsic behavior of a single forward generation.

## 3. Learning Self-Correction from Paired Rollouts

### 3.1. The Challenge: Self-Correction Signals Are Sparse

Teaching self-correction with RL requires learning signals that explicitly demonstrate how incorrect responses can be revised into correct ones, i.e., rollouts of the form *wrong → correct*. However, standard RL relies solely on

*Table 1.* $acc_{@1}$ and $acc_{@2}$ denote the accuracy before and after self-correction, respectively. $\triangle^{c \to w}$ represents the proportion of cases where a correct answer is revised into a wrong one, $\triangle^{w \to c}$ denotes the proportion of cases where an incorrect answer is corrected.

| Methods | $acc_{@1}$ | $acc_{@2}$ | $\triangle^{c \to w} \downarrow$ | $\triangle^{w \to c} \uparrow$ |
|---|---|---|---|---|
| Qwen3-VL-8B-Thinking | 79.8 | - | - | - |
| + "Wait" | 79.8 | 79.9 | 0.2 | 0.3 |
| + "Alternatively" | 79.8 | 79.5 | 0.3 | 0.0 |

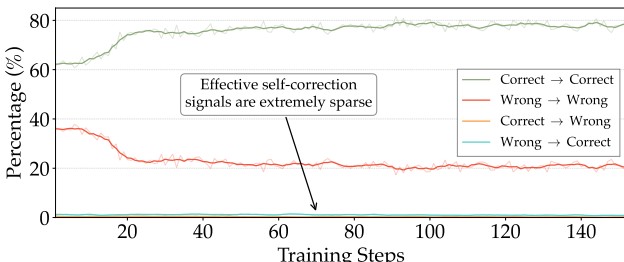

*Figure 2.* The percentage of different correction behaviors during RL training with a self-correction–encouraging prompt.

outcome-level rewards and does not explicitly encourage self-correction behavior. As a result, self-correction emerges only rarely and implicitly during training. Recent work attempts to amplify self-correction signals by prompting VLMs to generate self-correction rollouts (Wan et al., 2025). However, prompting alone cannot substantially alter model behavior, and effective self-correction remains rare.

We empirically quantify this sparsity. Under standard RL, even with manually appended "Wait"-style aha-moment tokens, only up to 0.3% of samples exhibit effective *wrong → correct* transitions (Table 1). Prompt-encouraged RL increases this fraction only marginally, to below 1% (Fig. 2). Moreover, the corresponding negative samples *correct → wrong* are also rare, indicating that the model collapses to simply maintaining the initial response. This extreme sparsity limits learning self-correction via RL.

### 3.2. Correction-Specific Rollout Augmentation

A key observation motivating our approach is that effective self-correction signals already exist in standard RL rollouts: for a given input, incorrect and correct responses often coexist within the same rollout group. By pairing them, we can explicitly construct samples that demonstrate effective correction behavior. Based on this, we propose **cor**re**cti**on–**s**pecific rollo**uts** (**Octopus**) augmentation.

A naive approach would directly pair responses from standard RL rollouts to form self-correction examples. However, these samples lie far outside the base VLM's distribution, leading to unstable RL training. To avoid this issue, we make the model generate rollouts in an explicit self-correction format (details in § 4.1), $\{(o_1^i \oplus \texttt{<sc>} \oplus o_2^i)\}_{i=1}^n$, where $o_1^i$ and $o_2^i$ denote the responses before and after the self-correction token. Importantly, neither $o_1^i$ nor $o_2^i$ is assumed to be correct or wrong. This step serves only to ensure that self-correction-style trajectories are in-distribution.

Given these rollouts, we construct augmented samples by recombining their components: we select $o_1$ from $\{o_1^i\}_{i=1}^n$ and $o_2$ from $\{o_2^i\}_{i=1}^n$, yielding $n^2$ paired rollouts in total. These pairs fall into 4 categories: *wrong → correct* (positive), *correct → correct* (positive), *correct → wrong* (negative), and *wrong → wrong* (negative). Among them, *wrong → correct* is the most informative, as it directly encodes effective self-correction behavior.

Assuming that $N$ rollouts are required for each policy update, we keep the $n$ originally generated rollouts to avoid relying entirely on offline data. We then select an additional $N - n$ samples from the augmented pool while balancing positive and negative examples. For positive examples, we prioritize *wrong → correct*, followed by *correct → correct* pairs when needed. For negative examples, we randomly sample from *correct → wrong* and *wrong → wrong*. In practice, we set $n = 8$ and $N = 16$, yielding 64 augmented rollouts, from which 16 balanced samples are selected. The complete procedure is summarized in Algorithm 1.

Octopus augmentation offers three key advantages: (i) it produces dense, explicit self-correction examples; (ii) it balances positive and negative samples, stabilizing RL optimization; (iii) since rollout generation is the most costly part of RL training, it substantially improves sample efficiency by reusing existing rollouts.

> **Takeaway for Octopus Augmentation**
>
> Octopus turns sparse, implicit signals into dense, explicit self-correction signals by pairing pre- and post-correction responses, improving sample efficiency and training stability without additional computational cost.

## 4. Training Recipe

Building on the key idea of Octopus, we present our complete training recipe. We begin with a cold-start stage that establishes the self-correction output format (§ 4.1). We then analyze the learning conflict between direct reasoning and self-correction under the RL objective (§ 4.2), and introduce a response-masking strategy to decouple these learning signals (§ 4.3).

### 4.1. Cold-Start and Data Construction

A straightforward way to induce the self-correction format $(o_1 \oplus \texttt{<sc>} \oplus o_2)$ is to prompt the model. However, prompting alone often yields $o_2$ responses that merely continue or partially revise $o_1$, resulting in incomplete post-correction reasoning. Pairing such $o_2$ with a different $o_1$ produces incoherent examples. To avoid this issue, we introduce a cold-start format-learning stage that ensures both $o_1$ and $o_2$

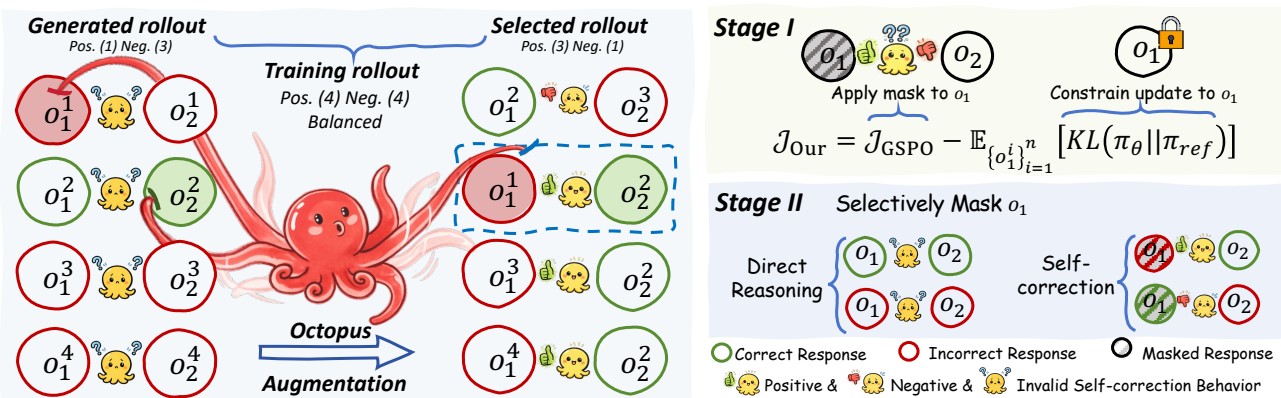

*Figure 3.* **Left**: Octopus augmentation pairs responses before and after the `<sc>` token to explicitly construct effective self-correction examples (wrong → correct), increasing their count from 0 to 4. It also produces an equal number of positive and negative samples (4 each), balancing the advantage distribution within each training group. **Right**: Our two-stage RL pipeline. In Stage I, we decouple self-correction learning by applying masks and KL regularization to $o_1$. In Stage II, we selectively unmask $o_1$ only for samples with non-conflicting reward signals, while keeping it masked for the remaining samples.

contain complete, self-contained reasoning. We consider two sampling strategies for constructing the SFT cold-start dataset: in-distribution sampling and mixed sampling.

**In-distribution Sampling.** This strategy samples all responses from the policy VLM $\pi_\theta$ and pairs them to form a self-correction format. For each input, we sample 4 responses. When all responses are correct, we select 4k instances to construct $o_1 \oplus \texttt{<sc>} \oplus o_2$. We use best-of-$N$ to select the best response as $o_2$, and randomly choose one of the remaining as $o_1$, ensuring that $o_2$ is better than $o_1$. When both correct and incorrect responses are present, we select 6k instances to construct $o_1 \oplus \texttt{<sc>} \oplus o_2$.

**Mixed Sampling.** In mixed sampling, responses before `<sc>` are sampled from the policy model $\pi_\theta$, while responses after `<sc>` are sampled from a stronger model $\pi_s$. We reuse the same 10k inputs $x$ and their corresponding $o_1$ from in-distribution sampling. To obtain higher-quality corrections, we generate $o_2$ using $\pi_s$, conditioned on the input $x$, the ground truth, and $o_1$: $o_2 \sim \pi_s(\cdot \mid x, o_1, \mathrm{gt})$.

**Setup.** We adopt Qwen3-VL-8B-Instruct (Yang et al., 2025) as $\pi_\theta$ and its larger variant, Qwen3-VL-30B-A3B-Instruct, as $\pi_s$. Since the cold-start stage mainly serves to learn the self-correction format and initialize RL training, we select the sampling strategy based on downstream RL performance conducted on ViRL-39k (Wang et al., 2025a).

**Results.** Fig. 4 shows that in-distribution sampling induces a larger entropy drop than mixed sampling, resulting in overly low initial entropy that limits further improvement during RL. In contrast, mixed sampling maintains an entropy trajectory comparable to GSPO and achieves higher accuracy rewards than both GSPO and in-distribution sampling. These results demonstrate that self-correction format learning without entropy collapse is critical for RL training.

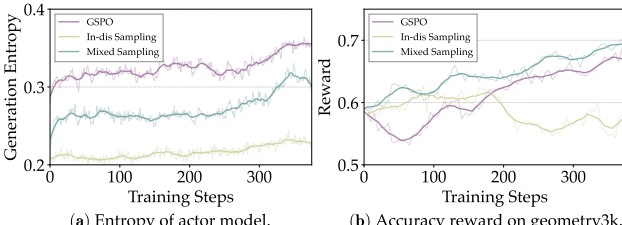

(a) Entropy of actor model.  (b) Accuracy reward on geometry3k.

*Figure 4.* Training dynamics of different methods. GSPO is initialized from the base $\pi_\theta$ and trained with standard RL. In-dis and Mixed Sampling are initialized from their corresponding SFT models and trained with Octopus RL strategy introduced in § 4.3.

---

**Takeaway for Cold-start Data Construction**

SFT-based cold-start is necessary for learning a self-correction format. Mixed sampling avoids entropy collapse and leads to more effective RL training.

---

## 4.2. Conflicts Between Direct Reasoning and Self-Correction in RL Training Objective

For RL training, the objective is to maximize the final accuracy reward. When responses contain self-correction, this objective can be achieved in two ways (Kumar et al., 2024): (i) producing the correct answer before `<sc>`, or (ii) correcting an initially incorrect response after `<sc>`. We refer to the former as *direct reasoning capability*, and the latter as *self-correction capability*. An ideal VLM should possess both abilities. However, when using a conventional binary reward that assigns 1 to correct final outcomes and 0 otherwise, these two learning signals are entangled. A natural approach to decouple these signals is reward shaping (Kumar et al., 2024; Wan et al., 2025). However, we find that under the single-pass self-correction setting, this strategy leads to reward hacking and unstable training.

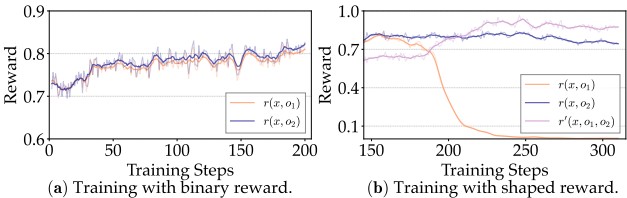

*(a)* Training with binary reward.  *(b)* Training with shaped reward.

*Figure 5.* Teaching self-correction with binary and shaped rewards. (a) Reward curves before and after self-correction under a binary reward setting, showing limited self-correction learning. (b) Reward curves with the shaped reward defined in Eq. (3), highlighting the emergence of reward hacking.

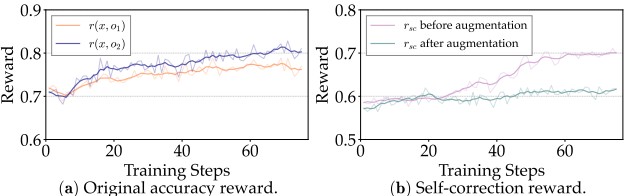

*(a)* Original accuracy reward.  *(b)* Self-correction reward.

*Figure 6.* Training curves for Stage I. (a) The reward gap between $o_1$ and $o_2$ gradually widens during training, indicating effective learning of self-correction. (b) The self-correction reward $r_{\text{sc}}$ (Eq. (6)) before and after Octopus augmentation. Octopus balances positive and negative samples, leading to stable reward dynamics.

**Setup.** We compare two reward designs: (i) a standard binary reward ($0/1$) based on the correctness of $o_2$, and (ii) a shaped reward proposed in Wan et al. (2025), defined as:

$$r'(x, o_1, o_2) = \begin{cases} 1.0 & \text{if } r(x, o_1) = 0, r(x, o_2) = 1 \\ 0.75 & \text{if } r(x, o_1) = 1, r(x, o_2) = 1 \\ 0.0 & \text{if } r(x, o_1) = 0, r(x, o_2) = 0 \\ -0.25 & \text{if } r(x, o_1) = 1, r(x, o_2) = 0 \end{cases}. \quad (3)$$

We initialize RL training from the cold-start model in § 4.1 and train it using GSPO (Zheng et al., 2025a) on the ViRL-39k dataset (Wang et al., 2025a). Octopus augmentation is disabled to isolate the effect of the RL objective.

**Results.** Fig. 5(a) shows that training with a binary reward fails to improve self-correction capability: the accuracy before and after `<sc>` remains nearly identical throughout training. The overlap of two curves during early iterations further indicates that SFT primarily serves as a format-learning stage and does not teach the model self-correction capability. Fig. 5(b) shows that reward shaping induces reward hacking after ∼200 training steps. As illustrated in Case 4 in the Appendix, the model deliberately produces an incorrect first response despite correct reasoning, followed by a trivial correction after `<sc>`. As reflected by the second-response accuracy and the shaped self-correction reward curves, this behavior induces training instability and ultimately degrades the model's overall reasoning capability.

> **Takeaway for Self-correction Reward**
>
> Jointly optimizing self-correction and direct reasoning limits self-correction learning. Reward shaping alone fails to decouple these signals and instead induces reward hacking and mode collapse.

### 4.3. Response-Masking Strategy for Decoupled Learning

To address the challenge of entangled learning objectives in single-pass RL, we decouple self-correction and direct reasoning through a two-stage training framework based on a response-masking strategy.

**Stage I: Learning Self-Correction Only.** To learn effective self-correction without reward hacking, we focus only on self-correction in the early stage of RL. Specifically, we treat the pre-correction response $o_1$ as fixed context: loss is masked for all tokens in $o_1$ and the policy is updated solely based on the post-correction response $o_2$. Additionally, we apply a KL loss on $o_1$, constraining its distribution to the reference model. The resulting optimization objective is:

$$\mathcal{J}_{\text{stage I}} = \mathcal{J}_{\text{GSPO}} - \mathbb{E}_{\{o_1^i\}_{i=1}^G} \left[ KL(\pi_\theta \,\|\, \pi_{\text{ref}}) \right]. \quad (4)$$

The importance sampling ratio used in GSPO is only calculated based on $o_2$:

$$s_i(\theta) = \frac{\pi_\theta(o_2 \mid x, o_1 \oplus \texttt{<sc>})}{\pi_{\text{old}}(o_2 \mid x, o_1 \oplus \texttt{<sc>})}. \quad (5)$$

The rule-based reward is defined as:

$$\begin{aligned} r_{\text{f}}(x, o_1, o_2) &= \min(r_{\text{f}}(x, o_1), r_{\text{f}}(x, o_2)) \\ r_{\text{sc}}(x, o_1, o_2) &= 0.9 \cdot r'(x, o_1, o_2) + 0.1 \cdot r_{\text{f}}(x, o_1, o_2), \end{aligned} \quad (6)$$

where $r_{\text{f}}$ is the format reward, and $r'$ is the shaped reward defined in Eq. (3) to strengthen self-correction behavior.

We report the training curves in Fig. 6 using the above objective with Octopus augmentation. Fig. 6(a) shows that the accuracy gap between $o_2$ and $o_1$ (measured on original, non-augmented rollouts) gradually widens over training, indicating that Stage I successfully improves self-correction capability. Fig. 6(b) shows that the self-correction reward $r_{\text{sc}}$ with Octopus augmentation remains stable, as augmentation balances positive and negative samples. In contrast, without Octopus augmentation, $r_{\text{sc}}$ exhibits a sharp increase during training, caused by an increasing dominance of positive samples in rollout groups. Such an imbalanced advantage distribution can weaken the effective learning signal during RL training (Wang et al., 2025a; Liu et al., 2025).

**Stage II: Co-evolving Reasoning and Correction.** The model's direct reasoning capability determines the starting

point for self-correction. In Stage II, we jointly improve both reasoning and self-correction by unmasking $o_1$ in the objective of Stage I and removing the KL term in Eq. (4). However, naively unmasking $o_1$ for all samples can introduce conflicting training signals for samples with effective self-correction signals, e.g., $(o_1 \oplus \texttt{<sc>} \oplus o_2)$. Because such samples receive positive rewards, backpropagating through $o_1$ would incorrectly reinforce a wrong direct response. In contrast, samples whose correctness remains unchanged before and after $\texttt{<sc>}$ (i.e., *correct → correct* or *wrong → wrong*) mainly provide learning signals for direct reasoning and do not suffer from this conflict. Therefore, we apply *selective unmasking*: $o_1$ is unmasked only for samples with consistent correctness before and after $\texttt{<sc>}$. This design enables jointly training of both capabilities, effectively preventing gradient conflicts induced by mixed optimization.

---

**Takeaway for Two-Stage RL Training**

Stage I isolates self-correction learning by treating $o_1$ as fixed context and updating the policy only from $o_2$. Stage II selectively unmasks $o_1$ when reward signals are non-conflicting, co-evolving both direct reasoning and self-correction.

---

# 5. Experiments

## 5.1. Setup

**Implementation Details.** For SFT cold-start, we apply the mixed sampling strategy to the LLaVA-CoT dataset (Xu et al., 2025) (§ 4.1), yielding 10k self-correction–formatted samples. For RL training, we perform the proposed RL training on ViRL-39k (Wang et al., 2025a). To handle off-policy signals introduced by augmentation, we adopt GSPO (Zheng et al., 2025a) without the online filter. All training is conducted on 8 NVIDIA H100 GPUs. More implementation details are provided in Appendix A.

**Baselines.** We compare our method against 3 categories of baselines. (i) **Closed-source VLMs**: GPT-4o (Hurst et al., 2024), OpenAI-o1 (Jaech et al., 2024), and Claude-3.7-Sonnet (Anthropic, 2024). (ii) **Open-source reasoning VLMs around 8B scale**: MiMO-VL-7B (SFT and RL) (Xiaomi, 2025), InternVL3.5-8B-RL (Wang et al., 2025b), and Qwen3-VL-8B-Thinking (Yang et al., 2025). (iii) **RLVR and self-correction baselines**: we reproduce GRPO (Shao et al., 2024), DAPO (Yu et al., 2025), GSPO (Zheng et al., 2025a), and SRPO (Wan et al., 2025) using the same dataset on Qwen3-VL-8B-Instruct (Yang et al., 2025).

**Benchmarks.** We select 2 types of benchmarks to evaluate our model: (i) **Math-related**: MathVista (Lu et al., 2023), MathVerse (Zhang et al., 2024a), and WeMath (Qiao et al., 2025). (ii) **General Task**: HallusionBench (Guan et al., 2024), MMStar (Chen et al., 2024), MMMU (Yue et al.,

2024), and CharXiv (Wang et al., 2024b). Evaluation is conducted using VLMEvalKit (Duan et al., 2024).

## 5.2. Main Results

**Performance.** Table 2 presents the performance of our Octopus model across 7 benchmarks. Octopus consistently and substantially improves over the base model Qwen3-VL-8B-Instruct, achieving an average accuracy gain of 9.5 points. Moreover, except on MathVerse, Octopus outperforms Qwen3-VL-8B-Thinking, the officially released reasoning-enhanced variant of the same backbone. Among RLVR baselines, we observe that GSPO, which uses sequence-level importance sampling, consistently outperforms GRPO and DAPO, both of which rely on token-level objectives. Building on GSPO, Octopus further improves reasoning performance through explicit self-correction, yielding an additional 1.0 average accuracy gain. Compared to the self-correction baseline SRPO, Octopus augmentation significantly increases effective self-correction reward signals and achieves better performance across all evaluated tasks. Overall, Octopus establishes a new state of the art among open-source VLMs of comparable size by explicitly and efficiently learning self-correction.

**Efficiency.** Table 2 compares training efficiency and average accuracy of different RLVR methods under varying numbers of rollout samples. We report the average per-step time cost over the first 10 training steps. Across RLVR baselines, increasing the number of rollout samples significantly improves accuracy, but at the cost of roughly doubling the training time per step. In contrast, Octopus leverages rollout augmentation to increase the number of rollouts per input from 8 to 16 *without any additional cost*. Since rollout generation is one of the most expensive components in RL training, this substantially reduces overall training time. As a result, our method achieves higher accuracy than GSPO with $n = 16$ rollouts while using only $0.72\times$ the training time. Its training time is comparable to baselines with $n = 8$, with the slight overhead attributed to updating the policy with a larger rollout set. These results demonstrate that by exploiting the self-correction structure, Octopus significantly accelerates RL training while simultaneously improving reasoning performance. We also provide inference latency and output token numbers in Appendix B.2 to demonstrate the inference efficiency of Octopus further.

## 5.3. Ablation Study

To understand the contributions of each component in Octopus, we conduct ablation studies along two dimensions: (i) Octopus augmentation and (ii) training strategy.

**Ablation on Octopus augmentation.** We first analyze the impact of the most critical component, Octopus augmentation. As shown in the second-to-last row of Table 3, re-

*Table 2.* Comparison between our model and baselines of similar scale across 7 benchmarks. The best and second-best results among open-source models are highlighted in **bold** and underline, respectively. *Gen.* and *Total* denote the rollout generation and the total training time per step (in seconds). `Octopus-8B` generates 8 rollouts during inference and augments them to 16 during training.

| Model | Time | | Math-Related | | | General Task | | | | |
|---|---|---|---|---|---|---|---|---|---|---|
| | Gen. | Total | MathVista | MathVerse | WeMath | HallBench | MMStar | MMMU$_{val}$ | CharXiv$_{RQ}$ | Avg. |
| *Closed-source VLMs* | | | | | | | | | | |
| GPT-4o | - | - | 63.8 | 50.2 | 68.8 | 56.2 | 64.7 | 69.1 | 50.5 | 60.5 |
| Claude-3.7-Sonnet | - | - | 66.8 | 52.0 | 72.6 | 55.4 | 65.1 | 71.0 | 64.2 | 63.9 |
| OpenAI-o1 | - | - | 73.9 | 57.0 | 98.7 | - | - | 78.2 | 55.1 | - |
| *Open-source Reasoning VLMs* | | | | | | | | | | |
| MiMo-VL-7B-SFT | - | - | 81.8 | 61.4 | 77.7 | 62.1 | 70.0 | 64.6 | 54.8 | 67.5 |
| MiMo-VL-7B-RL | - | - | 81.5 | 61.0 | 76.1 | 63.5 | 73.7 | 66.7 | 53.2 | 68.0 |
| InternVL3.5-8B-RL | - | - | 74.2 | 61.5 | 65.8 | 54.5 | 69.3 | 71.2 | 44.4 | 63.0 |
| Qwen3-VL-8B-Thinking | - | - | 79.8 | **69.2** | 83.0 | 62.7 | 74.1 | 71.8 | 53.0 | 70.5 |
| *Base VLM and RLVR baselines* | | | | | | | | | | |
| Qwen3-VL-8B-Instruct | - | - | 76.9 | 52.6 | 70.5 | 58.8 | 69.7 | 62.0 | 45.1 | 62.2 |
| + DAPO (rollout.n=16) | - | - | 80.3 | 68.5 | 82.4 | 63.7 | 74.7 | 71.4 | 52.8 | 70.5 |
| + GRPO (rollout.n=8) | 364.9 | 845.1 | 79.1 | 66.0 | 76.2 | 61.6 | 72.1 | 70.4 | 50.7 | 68.0 |
| + GRPO (rollout.n=16) | 679.1 | 1428.4 | 80.8 | 66.3 | 78.5 | 62.8 | 73.1 | 70.6 | 51.4 | 69.1 |
| + GSPO (rollout.n=8) | 361.7 | 753.1 | 78.5 | 63.7 | 74.0 | 62.3 | 72.5 | 67.4 | 47.9 | 66.6 |
| + GSPO (rollout.n=16) | 688.9 | 1322.8 | 81.0 | 68.4 | **84.0** | 62.5 | 72.7 | 70.8 | 55.3 | 70.7 |
| + SRPO (rollout.n=8) | 410.9 | 895.8 | 77.9 | 64.1 | 75.8 | 60.8 | 72.9 | 69.4 | 51.2 | 67.4 |
| + SRPO (rollout.n=16) | 816.6 | 1543.3 | 79.8 | 64.2 | 76.9 | 61.2 | 73.3 | 69.7 | 52.7 | 68.3 |
| `Octopus-8B` (Ours) | 344.7 | 958.1 | **82.1** | 68.5 | **84.0** | **64.2** | **75.2** | **72.4** | **55.7** | **71.7** |

*Table 3.* Ablation studies on Octopus augmentation and training strategy.

| Setting | **MathVista** | **MathVerse** | **WeMath** | **HallBench** | **MMStar** | **MMMU**$_{val}$ | **CharXiv**$_{RQ}$ | **Avg** |
|---|---|---|---|---|---|---|---|---|
| Qwen3-VL-8B (Base) | 76.9 | 52.6 | 70.5 | 58.8 | 69.7 | 62.0 | 45.1 | 62.2 |
| Octopus-8B (Ours) | 82.1 | 68.5 | 84.0 | 64.2 | 75.2 | 72.4 | 55.7 | 71.7 |
| w/o RL (SFT only) | 77.3 | 58.7 | 70.1 | 62.0 | 69.7 | 58.0 | 48.3 | 63.4$^{\downarrow 8.3}$ |
| w/o Stage I | 81.2 | 67.4 | 79.5 | 63.0 | 73.9 | 69.6 | 53.9 | 69.8$^{\downarrow 1.9}$ |
| w/o Octopus Augmentation | 79.0 | 63.9 | 75.7 | 62.0 | 71.7 | 68.8 | 50.5 | 67.4$^{\downarrow 4.3}$ |
| w/ Randomly Augmentation | 80.0 | 64.8 | 78.2 | 62.3 | 72.4 | 69.3 | 52.9 | 68.6$^{\downarrow 3.1}$ |

moving augmentation yields performance similar to RLVR baselines trained with fewer rollout samples, leading to a substantial drop in reasoning capability. To dive into whether the gains from Octopus augmentation stem from merely increasing the number of training samples or from enriching effective self-correction signals, we report in the last row of Table 3 the results of a random augmentation, where samples are concatenated at random instead of following the rules in Section 3.2. Random augmentation provides only a slight improvement over no augmentation (67.4 to 68.6) and remains far (↓ 3.1) from `Octopus-8B`. This gap demonstrates that performance gains are driven by enriching *effective self-correction signals*, rather than simply increasing rollout size.

**Ablation on Training Strategy.** To examine the necessity of each component in the training framework, we report in Table 3 the results of the SFT cold-start model and a variant trained with only Stage II during RL. The results show that, compared to the base model, SFT alone yields only marginal improvement, increasing accuracy from 62.2 to 63.4, and remains substantially below the performance of `Octopus-8B` (71.7). This indicates that within the Octopus framework, SFT primarily serves to learn the self-correction format rather than to deliver generalizable capability gains. Moreover, removing Stage I during RL training leads to a 1.9-point drop in accuracy. This confirms the critical role of Stage I in decoupling self-correction learning and further demonstrates that strengthening self-correction capability is essential for improving overall reasoning performance.

### 5.4. More Results and Analysis

**Self-correction Performance and Test-Time Scaling.** To evaluate whether self-correction is truly learned as a capability that improves reasoning performance, we report the performance of `Octopus-8B` before and after self-correction, as well as its test-time scaling (TTS) behavior by appending additional `<sc>` tokens to trigger further correction. As

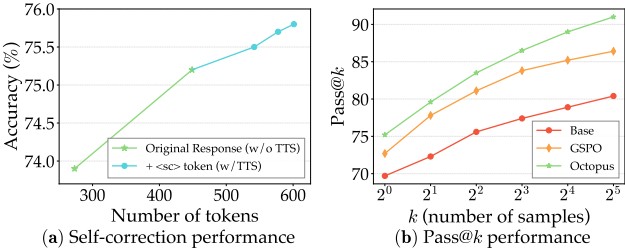

(a) Self-correction performance

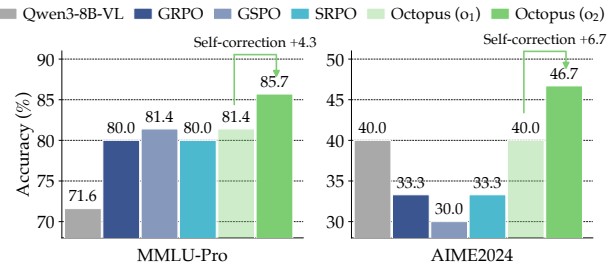

(b) Pass@k performance

*Figure 7.* Test-time scaling (TTS) performance on MMStar. **Left**: Sequential TTS achieved by appending <sc> tokens to trigger self-correction. Green points denote the original performance without TTS, and blue points indicate responses with TTS triggers. The x-axis shows the average cumulative number of tokens during inference. **Right**: Comparison of pass@$k$ performance.

*Figure 8.* Performance compared with different baselines on text-only input tasks. (Left: MMLU-Pro (Wang et al., 2024a), Right: AIME2024 (AI-MO, 2024)). The $o_1$ and $o_2$ variants correspond to the evaluation results of Octopus w/o and w/ self-correction, respectively.

illustrated in Fig. 7(a), the green curve shows that responses generated after the <sc> token achieve both higher accuracy and better token efficiency than those generated before <sc>. Moreover, although the model is trained to perform only a single round of self-correction, the controllable <sc> token allows us to explicitly trigger additional correction steps at inference time. The blue curve demonstrates that forcing the model to perform further self-correction via TTS progressively improves both accuracy and inference token efficiency, validating that self-correction is a generalizable and scalable capability.

**Pass@$k$ Performance.** Pass@$k$ is a crucial metric for evaluating a model's potential to solve a question within $k$ attempts, and is often regarded as a proxy for the model's reasoning boundary (Brown et al., 2024; Yue et al., 2025). We report the pass@$k$ accuracy on MMStar in Fig. 7(b). As $k$ increases, the performance margin between `Octopus-8B` and the GSPO baseline becomes more pronounced, increasing from 2.5 at pass@1 to 4.6 at pass@32. Moreover, compared to both GSPO and the base model, the larger performance margin achieved by Octopus further indicates that its reasoning boundary is substantially extended by the learned self-correction capability. We attribute this improvement to the augmented signals introduced by Octopus during training, which encourage the model to explore beyond its original distribution. This effect helps maintain higher entropy throughout training, leading to a stronger and more robust reasoning boundary.

**Performance on Text-only Domain.** Although Octopus is trained only on data with vision-language inputs, it naturally generalizes beyond multimodal settings. As shown in Fig. 8, Octopus-8B outperforms the baseline on text-only benchmarks such as AIME2024 (AI-MO, 2024) and MMLU-Pro (Wang et al., 2024a), whereas the baselines fail to improve and even degrade text-only performance. This suggests that self-correction is not confined to multimodal reasoning, but instead reflects a modality-agnostic strategy for improving model capability.

# 6. Related Works

**VLM Reasoning.** Reasoning capability is central to solving complex tasks with large language models (Jaech et al., 2024; Guo et al., 2025). The success of text-based reasoning has also driven the exploration of reasoning with multimodal data (Xu et al., 2025). Two primary paradigms are commonly used to enhance VLM reasoning: supervised fine-tuning (SFT) and reinforcement learning (RL). SFT requires reasoning trajectories with chain-of-thoughts (CoT) (Wei et al., 2022), which are often distilled from more powerful VLMs. RL methods, on the other hand, actively explore diverse reasoning trajectories and only require outcome-level rewards (Zhang et al., 2025; Peng et al., 2025; Wang et al., 2025a;c). While effective, RL typically requires a large number of rollout generations, which is time-consuming, and also suffers from advantage vanishing within training groups (Wang et al., 2025a). Prior work (Liu et al., 2025) explores rollout modification by adding noise to input images to encourage exploration. However, this approach is not designed to enrich reasoning or self-correction signals and incurs additional inference cost.

**Self-Correction.** Self-correction has been shown to improve reasoning performance across a range of tasks (Madaan et al., 2023; Zhang et al., 2024b). To enable controllable self-correction at test time, prior work typically relies on multi-pass prompting with explicitly designed correction formats (Kumar et al., 2024; Zeng et al., 2025; Ding & Zhang, 2025). However, such approaches require extensive model interaction and long contextual histories, making them token-inefficient at inference. The work most closely related to ours is SRPO (Wan et al., 2025), which studies single-pass self-correction in RL. SRPO uses prompting and reward shaping to encourage reflective behavior, but effective self-correction examples must still emerge spontaneously during rollouts, resulting in sparse learning signals. Moreover, self-correction and direct reasoning are jointly optimized, which can lead to mode collapse.

# 7. Conclusion

In this paper, we investigate how to improve reasoning performance by learning self-correction as a controllable behavior via RL. We propose Octopus, a rollout augmentation framework that constructs dense, explicit self-correction examples by pairing responses within rollout groups, enriching learning signals without additional generation cost. Moreover, we decouple the learning of self-correction and direct reasoning by response masking, avoiding objective conflicts and enabling both capabilities to be learned. Extensive experiments across seven benchmarks demonstrate that `Octopus-8B` achieves the best performance in both direct generation and test-time scaling, while requiring only $0.72\times$ training time per step compared to the best baseline.

These results highlight self-correction as a crucial capability for VLMs and show that explicitly learning it leads to more capable, efficient, and robust VLM reasoning. More broadly, our findings suggest that synthesizing structured supervision from policy samples is a promising direction for improving performance and reducing training cost.

## Acknowledgements

This research is supported in part by NSF IIS-2508145 and Amazon Research Award.

## Impact Statement

The goal of our paper is to improve the reasoning performance of VLMs. There is no specific social impact we need to highlight and clarify in this section.

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

# A. Implementation Details

## A.1. Training Details

In this section, we describe the training details of the different methods. We implement our SFT on the LLaMA-Factory (Zheng et al., 2024) and RL on the Easy-R1 (Zheng et al., 2025b) framework. All RL experiments in this paper are conducted on the ViRL-39k dataset (Wang et al., 2025a). We follow the experimental setup of Easy-R1 (Zheng et al., 2025b), and use Geometry-3k (Lu et al., 2021) as the validation set to select the best training checkpoint.

*Table 4.* Detailed training hyperparameters for different methods.

| Methods | RL-algo. | lr | Max Length | Warn-up Steps | Epoch |
|---|---|---|---|---|---|
| Qwen3-VL-8B-GRPO | GRPO | 1e-6 | 6144 | 10 | 5 |
| Qwen3-VL-8B-DAPO | DAPO | 1e-6 | 6144 | 10 | 5 |
| Qwen3-VL-8B-GSPO | GSPO | 1e-6 | 6144 | 10 | 5 |
| Qwen3-VL-8B-SRPO | GRPO | 1e-6 | 6144 | 10 | 5 |
| Octopus-SFT | - | 1e-6 | 8192 | - | 1 |
| Octopus-RL | GSPO | 1e-6 | 6144 | 10 | 5 |

We report the training hyperparameters used in our experiments in Table 4. For fair comparison, we fix the learning rate, max sequence length, warm-up steps, and number of training epochs across all methods. During training, we adopt vLLM (Kwon et al., 2023) as the inference backend, enabling faster rollout generation. The complete `<sc>` self-correction tokens we used is `\n\n<self-correction>\n</self-correction>\n\n`.

## A.2. Detailed Algorithm of Octopus Augmentation

In this section, we present the complete selection algorithm and selection rules for Octopus augmentation, as shown in Algorithm 1.

---

**Algorithm 1** Sample selection in Octopus

---

**Input:** training set size $N$, originical rollout set $\mathcal{S} = \{(o_1^i \oplus \texttt{<sc>} \oplus o_2^i)\}_{i=1}^n$
**Output:** Training set $\mathcal{D}$
Initialize buffer $\mathcal{B} \leftarrow \emptyset$.
**for** $i = 1$ **to** $N$ **do**
  **for** $j = 1$ **to** $N$, $j \neq i$ **do**
    $\mathcal{B} \leftarrow \mathcal{B} \cup (o_1^i \oplus \texttt{<sc>} \oplus o_2^j)$
  **end for**
**end for**
Initialize $\mathcal{D} \leftarrow \mathcal{S}$.

**Pre-defined rules** $\mathcal{R}$:
  1. Compute the correctness of each sample in $\mathcal{S}$. Calculate the number of correct and wrong samples as $n_c$, and $n_w$, respectively.
  2. If $|n_c| = |\mathcal{S}|$ or $|n_w| = |\mathcal{S}|$, randomly select $b_{ij}$ to fill $\mathcal{D}$, since the advantage vanishes in this case.
  3. Sample $N/2| - |n_c|$ examples with *wrong→correct* signal. If fewer than $N/2| - |n_c|$ samples are available, we supplement them with *correct→correct*. Then, we select negative samples from *correct→wrong* and *wrong→wrong* to fill the training set.

**repeat**
  Select a sample $b_{ij}$ based on pre-defined rules $\mathcal{R}$
  $\mathcal{D} \leftarrow \mathcal{D} \cup b_{ij}$
**until** $|\mathcal{D}| = N$
**return** $\mathcal{D}$

---

## A.3. Inference Prompts

In this section, we present the reasoning prompts used during training and evaluation for different methods. For the vanilla RLVR baselines, we follow the settings of Easy-R1 (Zheng et al., 2025b) and prompt the models to first reason through the

problem and then provide the final answer within a designated box.

---
**Prompt for GRPO DAPO and GSPO**

{Question}

You first think through the reasoning process as an internal monologue, enclosed within `<think>` `</think>` tags. Then, provide your final answer enclosed within \boxed{}.

---

For SRPO (Wan et al., 2025), we follow their RL training settings and use the following prompt:

---
**Prompt for SRPO**

{Question}

Solve the user's question step by step. First, think about the reasoning process internally and write it inside `<think>` and `</think>` tags. Then provide the first answer in LaTeX format, wrapped with $...$, and the final result must use \boxed{}. Wrap this answer inside `<answer>` and `</answer>` tags. After that, perform a critical self-reflection on the previous reasoning and answer, writing the reflection inside `<reflection>` and `</reflection>` tags. Then, based on the reflection, generate a new reasoning process and a new answer: the new reasoning is again inside `<think>` and `</think>`, and the new answer is inside `<answer>` and `</answer>`, still using LaTeX $...$ and \boxed{}. Make sure both reasoning steps are clear and detailed. Even if the final answer does not change, the second reasoning must incorporate improvements based on the reflection. Always strictly follow the sequence: `<think>`...`</think>` `<answer>`...`</answer>` `<reflection>`...`</reflection>` `<think>`...`</think>` `<answer>`...`</answer>`. Example: `<think>` Since $1+1=2$, so the answer is $2$. `</think>``<answer>` The answer is $\boxed{2}$. `</answer>``<reflection>` The reasoning is correct but too brief; I could have explained the addition more explicitly. `</reflection>``<think>` Adding $1$ and $1$ together results in $2$ because $1$ plus $1$ means taking one and adding another one, leading to $2$. `</think>``<answer>` The answer is $\boxed{2}$. `</answer>`. All reasoning, answer, and reflection steps must be included without omission.

---

For Octopus, we prompt the model to explicitly generate responses in the $o_1 \oplus$ `<sc>` $\oplus\ o_2$ format. The detailed prompt we use is as follows:

---
**Prompt for Octopus-8B**

{Question}

You first think through your reasoning process as an internal monologue, enclosed within `<think>` `</think>` tags. Then, provide your final answer enclosed within \boxed{}. If you believe the answer can be further enhanced, generate `<self-correction>` `</self-correction>` tags enclosed with no content, and regenerate a new reasoning process and a new answer from scratch after that. The new response should first think through your reasoning process as an internal monologue, enclosed within `<think>` `</think>` tags. Then, provide your final answer enclosed within \boxed{}. All reasoning, answer steps must be included without omission.

---

We also report the prompt we used during the cold-start data construction stage:

---
**Prompt for Cold-Start Mixed Sampling**

Question: {question}
Ground Truth: {ground_truth}
Reference Answer: {reference_answer}
Instruction:
Based on the provided information, generate a new, complete answer.
- Ensure the reasoning is correct and leads to the Ground Truth.
- If the Reference Answer is wrong, correct it implicitly by providing the right derivation.
- If the Reference Answer is correct, rewrite it to be clearer and more logical while keeping a similar length.

---

> - You first think through the reasoning process as an internal monologue, enclosed within `<think>` `</think>` tags. Then, provide your final answer enclosed within \boxed{}.
> - Do not generate thinking process that are too short.
> - Do not mention the Reference Answer in your response.

## B. Evaluation Details

### B.1. Evaluation Setting

We evaluate the performance of `Octopus-8B` across seven comprehensive benchmarks. All evaluations are conducted using the VLMEvalKit (Duan et al., 2024) framework, with vLLM (Kwon et al., 2023) serving as the inference backend. Detailed benchmark information is provided in Table 5.

For sampling, we set the temperature to $t = 0.6$, top-$p$ to 0.95, and top-$k$ to $-1$. For reasoning models, we cap the inference budget at 16,384 tokens, while instruct models are limited to 4,096 tokens. During answer evaluation, we extract responses enclosed in \boxed{} for reasoning models, and use the full generated outputs for instruct models.

*Table 5.* Detailed information of our evaluated benchmarks, *Evaluation Split* and *Reported Metric* are features in VLMEvalKit.

| Benchmark | Evaluation Split | Num of Sample | Reported Metric |
|---|---|---|---|
| MathVista (Lu et al., 2023) | `MathVista_MINI` | 1000 | `acc` |
| MathVerse (Zhang et al., 2024a) | `MathVerse_MINI` | 3940 | `mean(vision)` |
| WeMath (Qiao et al., 2025) | `WeMath` | 1740 | `Score (Loose)` |
| HallusionBench (Guan et al., 2024) | `HallusionBench` | 951 | `average` |
| MMStar (Chen et al., 2024) | `MMStar` | 1500 | `Overall` |
| MMMU (Yue et al., 2024) | `MMMU_DEV_VAL` | 900 | `validation overall` |
| CharXiv (Wang et al., 2024b) | `CharXiv_reasoning_val` | 1000 | `Overall` |

### B.2. Inference Efficiency

To evaluate the inference efficiency of Octopus-8B, we report the average reasoning latency and output token count on MathVista in Table 6, together with the baselines. The results show that although Octopus-8B naturally produces two responses, $o_1$ and $o_2$, within a single inference pass, the final output lengths of different methods converge to a similar range during RL training, leading to comparable inference latency overall. This indicates that Octopus-8B achieves stronger performance than the baselines under a similar inference token budget.

*Table 6.* Comparison of inference efficiency. The reported number is the average on MathVista.

| Methods | Inference Time (s) | Token Number | Accuracy |
|---|---|---|---|
| GRPO (rollout.n=8) | 163 | 1059 | 79.1 |
| GRPO (rollout.n=16) | 160 | 1012 | 80.8 |
| GSPO (rollout.n=8) | 167 | 1064 | 78.5 |
| GSPO (rollout.n=16) | 156 | 1041 | 81.0 |
| SRPO (rollout.n=8) | 133 | 879 | 77.9 |
| SRPO (rollout.n=16) | 148 | 930 | 79.8 |
| `Octopus-8B` | 166 | 1063 | 82.1 |

## C. Failure Attempts

We also explored an alternative strategy that randomly mixes elements from $o_1^i$ and $o_2^i$ to form augmented samples. However, this approach leads to training collapse in practice, as $o_1$ and $o_2$ follow different distributions. We therefore restrict each component to be sampled from its corresponding set, preserving the original response structure. We visualize the collapsed training curve in Fig. 9.

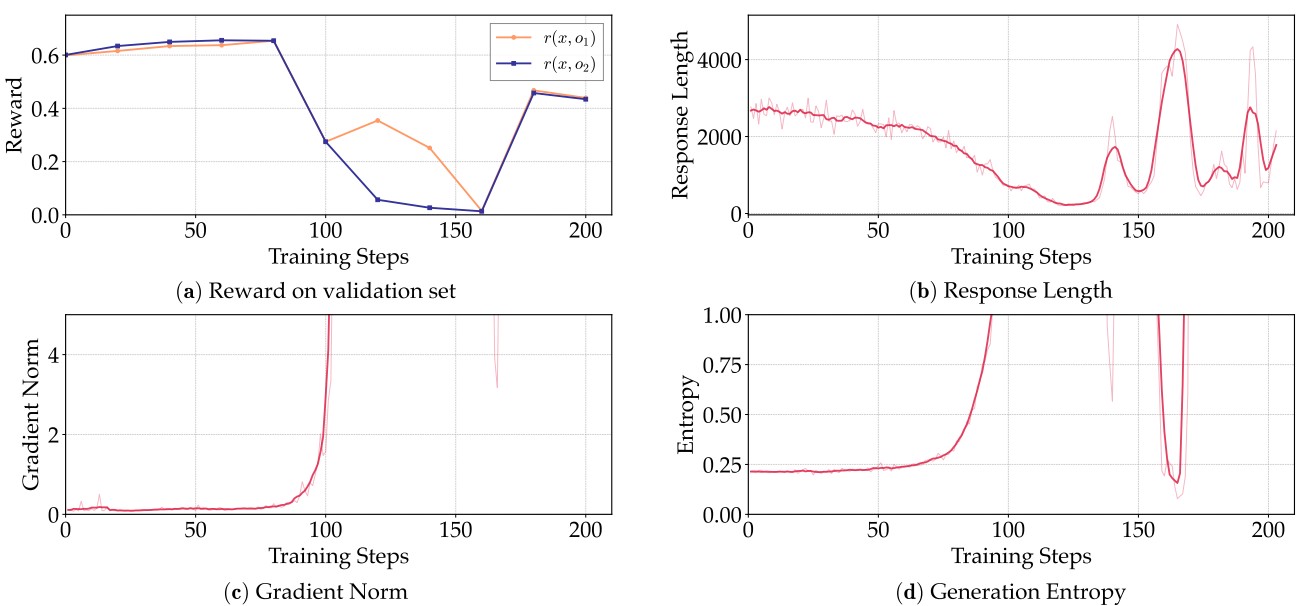

(a) Reward on validation set      (b) Response Length

(c) Gradient Norm      (d) Generation Entropy

*Figure 9.* Failure attempts. Training collapse during RL training.

## D. Case Study

In this section, we provide some case studies for `Octopus-8B`, demonstrating how `Octopus-8B` recovers errors in previous reasoning trajectories.

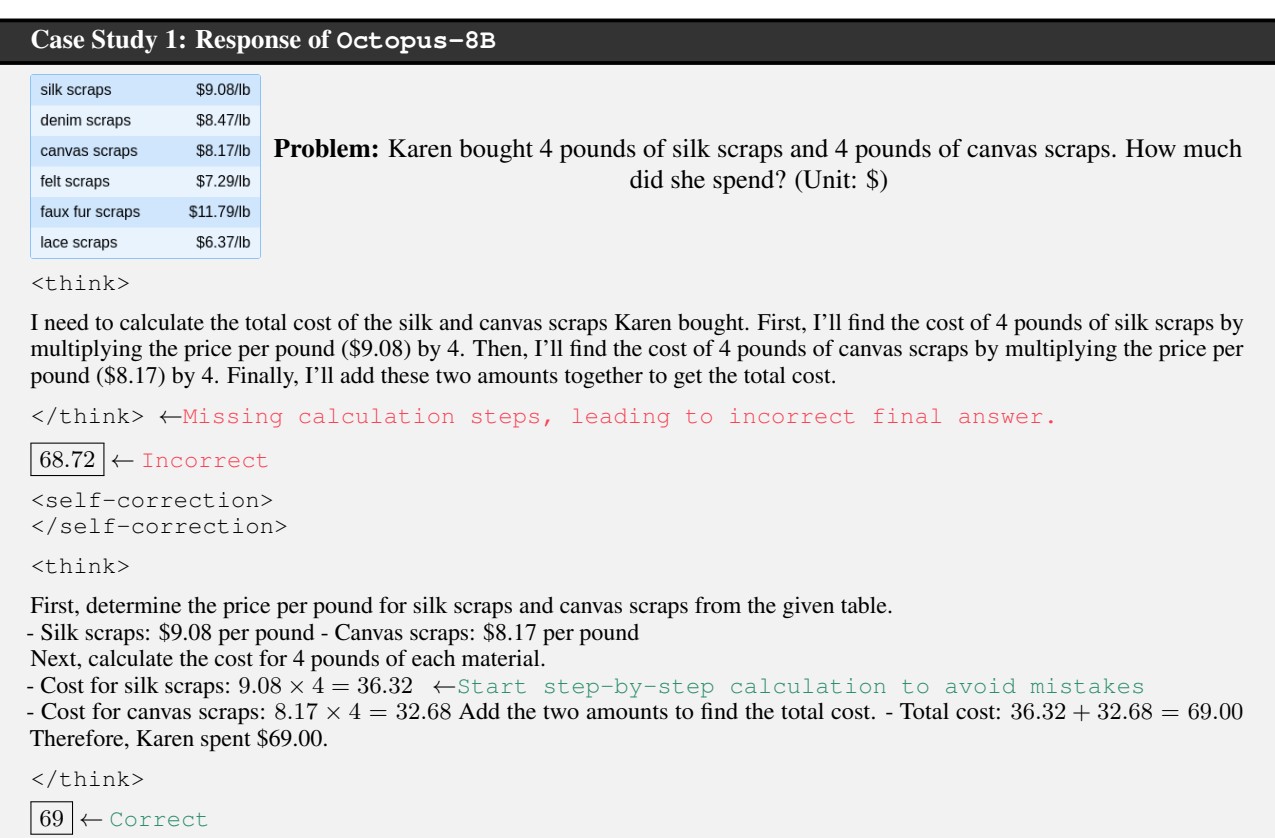

**Case Study 1: Response of `Octopus-8B`**

| | |
|---|---|
| silk scraps | $9.08/lb |
| denim scraps | $8.47/lb |
| canvas scraps | $8.17/lb |
| felt scraps | $7.29/lb |
| faux fur scraps | $11.79/lb |
| lace scraps | $6.37/lb |

**Problem:** Karen bought 4 pounds of silk scraps and 4 pounds of canvas scraps. How much did she spend? (Unit: $)

`<think>`

I need to calculate the total cost of the silk and canvas scraps Karen bought. First, I'll find the cost of 4 pounds of silk scraps by multiplying the price per pound ($9.08) by 4. Then, I'll find the cost of 4 pounds of canvas scraps by multiplying the price per pound ($8.17) by 4. Finally, I'll add these two amounts together to get the total cost.

`</think>` ←Missing calculation steps, leading to incorrect final answer.

68.72 ← Incorrect

`<self-correction>`
`</self-correction>`

`<think>`

First, determine the price per pound for silk scraps and canvas scraps from the given table.
- Silk scraps: $9.08 per pound - Canvas scraps: $8.17 per pound
Next, calculate the cost for 4 pounds of each material.
- Cost for silk scraps: $9.08 \times 4 = 36.32$ ←Start step-by-step calculation to avoid mistakes
- Cost for canvas scraps: $8.17 \times 4 = 32.68$ Add the two amounts to find the total cost. - Total cost: $36.32 + 32.68 = 69.00$
Therefore, Karen spent $69.00.

`</think>`

69 ← Correct

**Case Study 2: Response of `Octopus-8B`**

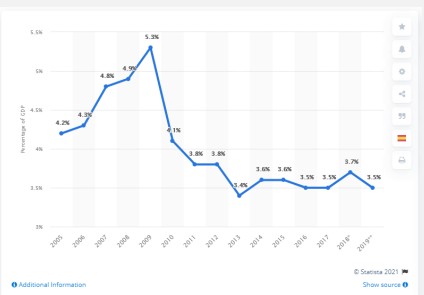

**Problem:** For how many years that the percentage value over 4?

`<think>`

I need to count the number of years where the percentage value is over 4. Looking at the line chart, I'll go year by year and check if the percentage exceeds 4%.

2005: 4.2% → over 4? Yes. 2006: 4.3% → over 4? Yes. 2007: 4.8% → over 4? Yes. 2008: 4.9% → over 4? Yes. 2009: 5.3% → over 4? Yes.
2010: 4.1% → over 4? No.   ←Incorrect reasoning step.
2011: 3.8% → over 4? No. 2012: 3.8% → over 4? No. 2013: 3.4% → over 4? No. 2014: 3.6% → over 4? No. 2015: 3.6% → over 4? No. 2016: 3.5% → over 4? No. 2017: 3.5% → over 4? No. 2018: 3.7% → over 4? No. 2019: 3.5% → over 4? No.

So, the years over 4% are 2005, 2006, 2007, 2008, and 2009. That's 5 years.

Wait, let me recount:  ←Self-reflection fail to identify mistakes.

2005: 4.2% → Yes 2006: 4.3% → Yes 2007: 4.8% → Yes 2008: 4.9% → Yes 2009: 5.3% → Yes

That's 5 years.

I think I missed nothing. The others are all below 4%.vspace5pt
Final count: 5 years.

`</think>`

5   ←Incorrect

`<self-correction>`
`</self-correction>`

`<think>`

To determine the number of years with a percentage value over 4, we need to review the data points for each year from 2005 to 2019 and count those where the value exceeds 4%.

- 2005: 4.2% →over 4 - 2006: 4.3% →over 4 - 2007: 4.8% →over 4 - 2008: 4.9% →over 4 - 2009: 5.3% →over 4
- 2010: 4.1% → over 4   ←Correct previous mistake in the second response

- 2011: 3.8% →not over 4 - 2012: 3.8% →not over 4 - 2013: 3.4% →not over 4 - 2014: 3.6% →not over 4 - 2015: 3.6% →not over 4 - 2016: 3.5% →not over 4 - 2017: 3.5% →not over 4 - 2018*: 3.7% →not over 4 - 2019**: 3.5% →not over 4

The years with percentages over 4% are 2005, 2006, 2007, 2008, 2009, and 2010, totaling 6 years.

The 2010 value is 4.1%, which is greater than 4%. I previously missed this in my initial count.  ←Even identify the mistakes in the first attempt

Therefore, the correct count is 6 years.

`</think>`

6   ←Correct

**Case Study 3: Response of `Octopus-8B`**

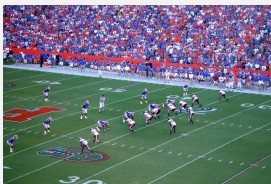

**Problem:** whats the lowest number yard line that you can see?

```
<think>
```

I need to identify the lowest numbered yard line visible in the image. Looking at the field, I can see the yard lines marked with numbers. Starting from the bottom of the image, I can clearly see the number "30" on the field. Moving up, I can see "40", "50", and "60" as well. The "1" is visible as part of the "1" on the 1-yard line, which is near the goal line. The "1" is at the very end of the field, so it should be the lowest numbered yard line. Therefore, the lowest number yard line I can see is 1. ←Overlooking visual content in the image

```
</think>
```

1  ←Incorrect

```
<self-correction>
</self-correction>
```

```
<think>
```

To determine the lowest numbered yard line visible, I need to examine the yard line markings on the field. The yard lines are clearly labeled with numbers, and I can see several of them. Starting from the bottom of the image (closest to the viewer), the visible numbers are: - 30 - 40 - 50 - 60 These are the primary yard lines marked on the field. The "1" is not visible in the image; it's located near the goal line, which is beyond the 0-yard line. ←Demonstrating 1 is not in the image However, the question asks for the lowest number yard line that can be seen. The lowest number among the visible markings is 30.

```
</think>
```

30  ←Correct

**Case Study 4: Reward hacking when training with shaped-reward**

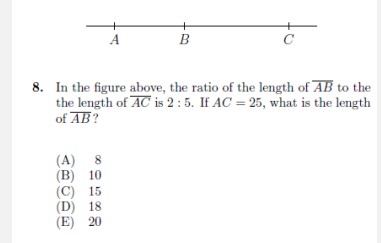

8. In the figure above, the ratio of the length of $\overline{AB}$ to the length of $\overline{AC}$ is 2 : 5. If $AC = 25$, what is the length of $\overline{AB}$?

(A)   8
(B)   10
(C)   15
(D)   18
(E)   20

**Problem:** In the figure above, the ratio of the length of line AB to the length of line AC is 2 : 5. If AC = 25, what is the length of line AB?

```
<think>
```

We are given that the ratio of AB to AC is 2:5, and AC = 25. We need to find the length of AB.
Since the ratio AB:AC = 2:5, this means that AB is 2 parts out of every 5 parts of AC.
So, if AC is 25, then each part is 25 divided by 5, which is 5.
Therefore, AB = 2 parts × 5 = 10.
Let me double-check:
If AB = 10 and AC = 25, then the ratio AB:AC = 10:25 = 2:5, which matches the given ratio.
So, the length of AB is 10.
The correct answer is (B) 10. ←Already generated correct response

```
</think>
```

$\boxed{D}$ ←Incorrect, reward hacking behavior

```
<self-correction>
</self-correction>
```

```
<think>
```

We are given that the ratio of the length of AB to AC is 2:5, and AC = 25. We need to find the length of AB.
Let's denote the length of AB as 2x and the length of AC as 5x, since their ratio is 2:5.
We are told that AC = 25, so we can set up the equation: 5x = 25
Solving for x: x = 25 ÷ 5 = 5
Now, substitute x = 5 into the expression for AB: AB = 2x = 2 × 5 = 10
Therefore, the length of AB is 10.
Checking the ratio: AB = 10, AC = 25, so the ratio is 10:25, which simplifies to 2:5, matching the given ratio.
Thus, the correct answer is (B) 10.

```
</think>
```

$\boxed{B}$ ←Correct

$r(x, o_1, o_2) = 1$ ← reward hacking

