# OpenReview forum: "Learning Self-Correction in Vision–Language Models via Rollout Augmentation"
_ICML.cc/2026/Conference — ICML 2026 regular_

### Official Review · Reviewer_NPWb · 2026-03-08

**Soundness:** 2
**Presentation:** 2
**Significance:** 3
**Originality:** 3
**Overall Recommendation:** 3
**Confidence:** 5

**Summary:**

This paper proposes a correction-specific rollout-augmentation framework that synthesizes dense self-correction signals during RL training. The key idea is to construct correct and incorrect reasoning trajectories and randomly pair them for self-correction trajectory, where cold-start and two stage RL are performed. To avoid decreasing direct reasoning ability, the authors update policy only based on trajectory after correction in stage 1, and unmask only correct trajectory before correction in stage 2. Experiments show effectiveness of the method.

**Compliance With Llm Reviewing Policy:**

Affirmed.

**Final Justification:**

After read the author response, the concerns are remain.
1. The concern about inconsistency is reflection point on error rather than linguistic expression. As in the authors' response that, "... each $o_2$ reasons from scratch rather than referencing specific errors in $o_1$", the random combination of $o_1$ and $o_2$ to form the self-correction samples may cause the inconsistent reflection points, and the authors haven't response to this.
2. The $o_1$ may not on-policy in RL training, and applying KL on such off-policy trajectories is not well justified. As described in sec.4.1, the authors construct 10k samples of $o_1\oplus$ \<sc\> $\oplus o_2$ for RL sampling. Although $o_1$ is initially generated by the policy model, subsequent RL updates has made it off-policy, and thus applying KL on $o_1$ may fail to regularize the distribution shift.
3. The inference of Octopus remains constrained to a fixed reasoning and correction pattern, and does not internalize the correction behavior into the model itself for automatic triggering.

Therefore, I will maintain my score.

**Key Questions For Authors:**

1. It is unclear whether it is fully on-policy training in RL or using off-policy samples, as presented in sec. 3.2. If using off-policy samples, why don't we perform fully on-policy RL;
2. Whether the tokens from $o_1$ in stage 1 receive gradients;
3. The $r_{sc}$ is not defined in Figure 6.

**Limitations:**

The authors haven't include limitations in their work.

**Strengths And Weaknesses:**

Strengths:
1. The authors construct the dense self-correction signals from simplicity multiple rollouts and combine the correct and incorrect ones for supervision;
2. The authors made efforts in avoiding decreasing direct reasoning ability when developing the self-reflection ability through two stage RL;
3. Experiments show the effectiveness of the method;

Weaknesses:
1. The reflection signals is concerned, as the authors randomly combine the incorrect and correct trajectories to form the self-correction samples, the reflection points may inconsistent;
2. The KL objective in stage 1 is doubted and should be validated, it seems like the incorrect behavior is injected into the model; Additionally, the effectiveness of stage 2 should be evaluated, as it is somewhat unsoundness;
3. The direct reasoning performance should be reported to validate whether the method obtain the performance gain by decrease the first attempt accuracy, and how frequently the Octopus trigger the correction behavior compared to other RL trained models, as well as the number of response tokens should be compared;
4. It is unclear whether the model truly learns general self-correction ability, e.g., whether models correct earlier mistakes more frequently? Whether they revise reasoning process more effectively?
5. The comparison with other self-correction methods should be presented;

---

> ### Author Rebuttal · Authors · 2026-03-31
>
> Thanks for your valuable feedback.
> > W1
> - **SFT ensures $o_1$ and $o_2$ reason from scratch, reducing inconsistency after concatenation**
> In Octopus, SFT trains both $o_1$ and $o_2$ as complete reasoning processes. Without it, RL on the base model often makes $o_2$ only partially revise $o_1$, leading to inconsistent concatenations and weaker training. After SFT, each response is self-contained, so concatenation better matches the rollout format and preserves augmentation quality.
> - **Ablation results validate the benefit of augmentation**
> Removing augmentation gives 67.4, close to the rollout=8 baseline and well below Octopus-8B (71.7). Even random augmentation reaches 68.6, outperforming the no-augmentation variant. This suggests augmentation does not introduce harmful inconsistency.
> ---
> > W2&Q2
> - **We do not inject wrong behavior**
> In Stage 1, the KL term on $o_1$ is used to keep its distribution close to the reference model, rather than introducing new errors. This design, similar to SCoRe [1], helps decouple the training objectives.
> - **$o_1$ receives gradients in Stage 1**
> The KL loss on $o_1$ is included in the GSPO objective, although we do not compute GSPO importance weights for tokens in $o_1$, as described in Eq.5.
> - **KL in Stage 1 benefits later RL**
> As shown in the table below, removing the KL term leaves $o_1$ performance unchanged but weakens $o_2$. We believe KL helps Stage 1 focus on learning self-correction, which provides stronger signals for Stage 2.
> - **Stage 2 is designed to jointly improve direct reasoning and self-correction capabilities**
> In Stage 1, we mainly learn the harder self-correction ability, while direct reasoning remains important as the basis for test-time scaling. Stage 2 therefore selectively unmasks $o_1$ only when no gradient conflict is introduced, avoiding reward hacking while improving both direct reasoning and self-correction. As shown below, Stage 2 significantly boosts direct reasoning and further improves $o_2$, indicating continued reasoning gains through self-correction.
> ||MathVista|Avg on 7 benchmarks|
> |-|-|-|
> |SRPO($o_1$)|79.6|68.1|
> |SRPO($o_2$)|79.8|68.3|
> |Octopus Stage1($o_1$)|78.9|66.4|
> |Octopus Stage1($o_2$)|80.8|69.2|
> |Octopus w/o KL($o_1$)|79.3|70.0|
> |Octopus w/o KL($o_2$)|81.0|70.9|
> |Octopus($o_1$)|79.3|70.2|
> |Octopus($o_2$)|82.1|71.7|
> ---
> > W3&W4
> - **Octopus doesn't weaken direct reasoning**
> As shown above, $o_1$ accuracy also improves. Stage 2 jointly improves direct reasoning and self-correction, and even after Stage 1, $o_1$ already outperforms the SFT model in Table 3. This suggests that learning self-correction can also benefit direct reasoning, consistent with prior work [1][4].
> - **Response length is well controlled**
> Although Octopus performs self-correction at inference, its response is only slightly longer than the baseline and gradually converges during training. Detailed results are provided in our response to Reviewer GBi9, W3&Q3.
> - **Octopus triggers more effective self-correction behavior**
> Like SRPO, Octopus performs exactly one round of self-correction, $o_1\oplus \<sc\>\oplus o_2$, but produces far more effective corrections (revises more incorrect o1 into correct o2). Across 7 benchmarks, the average gain from $o_1$ to $o_2$ is 1.5 for Octopus vs. 0.2 for SRPO. On MathVista, it corrects 3.7% of cases with an incorrect $o_1$, compared with only 0.3% for SRPO.
> - **Octopus can further conduct test-time scaling via self-correction**
> Although Octopus is trained with only 1 round of self-correction, Figure 7 shows that this ability naturally extends to test-time scaling: accuracy continues to improve as more \<sc\> tokens are appended ( $o_1=73.9,o_2=75.2,o_3=75.5,o_4=75.7,o_5=75.8$ )
> ---
> > W5
> - **We already compared with the most relevant baseline, SRPO**
> SRPO is the most recent and closest baseline, and to the best of our knowledge, the only on-policy RL method for VLM self-correction reasoning.
> - **Other self-correction baselines are not directly comparable**
> Methods such as Sherlock [4] and $R^3V$ [5] rely only on offline training (SFT or preference tuning) and were developed on older models such as Llama3.2. We tried reproducing them on stronger models like Qwen3-VL, but they performed poorly and could even underperform the base model, so we do not include them.
> We will reproduce any suggested baselines and discuss the results in the revision.
> ---
> > Q1
>
> In our 16-sample augmented rollout set, the original 8 samples are fully on-policy, while the other 8 mix $o_1$ and $o_2$ from different responses, introducing only a mild off-policy effect. In practice, this does not harm training stability. Octopus-8B also achieves much better pass@k than baselines, suggesting augmentation encourages exploration beyond the original policy distribution.
>
> ---
> > Q3
>
> $r_{sc}$ is the same as r' defined in Eq. 3. We will correct this typo in the revised version.
>
> # Reference
> Please see response to Reviewer fUWh

---

> > ### Author Rebuttal · Reviewer_NPWb · 2026-04-04
> >
> > Thanks for authors response. I believe some of my concerns have been addressed, however, the others remain.
> > 1. It is still fully unresolved that the reflection points may inconsistent when randomly combine the incorrect and correct trajectories to form the self-correction samples in stage 1 RL, and it is unclear whether the KL loss is applied on off-policy $o_1$, which is not soundness.
> > 2. It seems that the proposed method still requires a strict two-stage inference process, reasoning and correction, as in previous works, and does not appear to internalize the correction behavior into the model itself for automatic triggering.
> >
> > Based on these points, I will temporarily maintain my score.

---

> > > ### Author Response · Authors · 2026-04-04
> > >
> > > Thank you for your continued engagement and thoughtful feedback.
> > >
> > > ---
> > > > A1
> > > - **Our concatenation strategy does not introduce reflection inconsistency.**
> > >
> > >    A key design choice in Octopus is that self-correction is not a partial revision of the previous answer but the generation of a completely new, self-contained response. During the SFT stage, we explicitly teach the model to produce both $o_1$ and $o_2$ as full, independent reasoning chains. Because each $o_2$ reasons from scratch rather than referencing specific errors in $o_1$, concatenating an $o_1$ from one rollout with an $o_2$ from another will not cause inconsistency.
> > >
> > >    This design allows us to leverage existing RL rollouts to teach self-correction ability. It is also practical: in step-by-step reasoning, errors often accumulate over the course of generation. Under such conditions, partial reflection is usually insufficient to correct earlier mistakes and recover the correct answer [1]. We acknowledge that learning stronger partial reflection is an important direction. However, given the above-mentioned challenge, acquiring this ability efficiently is highly difficult. We leave it as future work.
> > > - **$o_1$ is fully on-policy in our RL training.**
> > >
> > >     In our augmented samples, $o_1$ is never off-policy. We first generate the original rollout set $\\{o_1^i\oplus\texttt{\<sc\>}\oplus o_2^i\\}_{i=1}^n$. Then, during concatenation, we select $o_1$ from $\\{o_1^i\\}^n\_{i=1}$ and $o_2$ from $\\{o_2^i\\}^n\_{i=1}$, forming new samples of the form $o_1\oplus\texttt{\<sc\>}\oplus o_2$. Here, $o_1$ is always fully generated on-policy and strictly follows the policy distribution. The only potential off-policy component is $o_2$ (now conditioned on a different $o_1$ prefix), since the purpose of augmentation is to encourage effective self-correction during training, i.e., revising an initially incorrect answer into a correct one. Only half of the sequences in our training rollout set are augmented; the other half are still fully on-policy original rollouts. Moreover, a small amount of off-policy signal in the sequence suffix can encourage the model to explore beyond its original distribution during RL training [2]. We also find that this design helps Octopus-8B achieve better pass@$k$ performance, outperforming GSPO by 5 points at pass@$32$ and the base model by around 10 points.
> > > - **The KL term in Stage 1 is used to focus learning on self-correction.**
> > >
> > >     Although we mask the $o_1$ tokens in the GSPO objective during Stage 1, the model update can still change the distribution of $\pi(\cdot \mid x)$ and indirectly alter the distribution over $o_1$. We apply KL regularization on $o_1$ against the reference model to avoid this drift, ensuring that learning focuses on improving $o_2$ quality. This helps decouple self-correction learning from direct reasoning, allowing the model to better learn how to generate improved $o_2$. A similar strategy was shown to be effective in prior work [3]. Our rebuttal experiments (in W2&Q2) also confirm that adding the KL loss in Stage 1 leads to better final $o_2$ accuracy after Stage 2 RL.
> > > ---
> > >
> > > > A2
> > >
> > > - **Octopus uses one-pass inference, with self-correction triggered inherently.**
> > > We want to clarify that Octopus does *not* require a separate two-stage inference pipeline. At inference time, we simply use the same prompt as in RL training, and the model automatically produces responses in the form of $o_1\oplus\texttt{\<sc\>}\oplus o_2$ in a single pass. The $\texttt{\<sc\>}$ token is produced by the model itself, not injected externally. The results for $o_1$ and $o_2$ reported in our paper are obtained by splitting this single-pass output at $\texttt{\<sc\>}$ for analysis purposes only. In the test-time scaling experiments, the model still generates $o_1$ and $o_2$ automatically; to further scale its reasoning, we find that manually appending additional $\texttt{\<sc\>}$ tokens can trigger further rounds of self-correction and lead to higher accuracy.
> > >
> > > - **Octopus triggers more effective self-correction then baselines**
> > >
> > >     We find that the most critical limitation of prior work is that, although the model may generate the self-correction format, it is rarely able to revise an incorrect answer into a correct one, which means the correction is superficial. Our experiments verify that, compared with the baseline method, which corrects wrong answers only $0.3\%$ of the time, Octopus significantly improves the correction success rate to $3.7\%$. Combined with the $0.72\times$ training time advantage, these results validate the effectiveness of our rollout augmentation strategy for creating more useful self-correction training samples, as well as our multi-stage training design.
> > > ---
> > >
> > > [1] Sherlock: Self-Correcting Reasoning in Vision-Language Models
> > >
> > > [2] Learning Autonomous Code Integration for Math Language Models
> > >
> > > [3] Training Language Models to Self-Correct via Reinforcement Learning

---

### Official Review · Reviewer_fUWh · 2026-03-12

**Soundness:** 3
**Presentation:** 4
**Significance:** 3
**Originality:** 3
**Overall Recommendation:** 4
**Confidence:** 4

**Summary:**

The paper proposes a self-correction specific reinforcement learning training technique that specifically trains vision language models to correct their reasoning chains. The results show that VLMs learn to self-correct their reasoning chains via reinforcement learning and improve performance on general and math VLM datasets. However, the method can be prone to reward hacking since the VLM can intentionally generate wrong reasoning path first then correct it to maximize reward.

**Compliance With Llm Reviewing Policy:**

Affirmed.

**Key Questions For Authors:**

-- A key question is about reward hacking. If the proposed algorithm will lead to reward hacking, why reward hacking does not appear in the first few steps of training but after 200 steps?

-- Since the base model is already instruct model, why do we still need SFT to perform cold start when a model can already follow instructions?

**Limitations:**

The paper lacks a limitation section, which can discuss reward hacking solution and compute efficiency issues.

**Strengths And Weaknesses:**

Strengths

-- The writing of the paper is clear and informative. Explanations and details are thorough.

-- The self-correction algorithm is novel, showing promising results at improving VLM visual reasonings.

Weaknesses

-- A main concern is the computation efficiency. For each input, the policy model is promoted to generate n^2 rollouts, compared to general GRPO n rollouts. This could double the cost of training to trade off a two percent accuracy. Can the authors report wall training time for each step or each 10 steps compared to the wall time of GRPO algorithm?

-- Another concern is the reward hacking the author discussed in the paper. Are there any methods or reward shaping that can reduce the risk of reward hacking? In a practical perspective, how do we know when reward hacking is expected to prevent over-training.

---

> ### Author Rebuttal · Authors · 2026-03-31
>
> We sincerely thank the reviewer for the thoughtful and thorough comments on our paper.
>
> ---
>
> > W1: Efficiency
>
> - Octopus only generates $n$ rollouts, then augmented to $n^2$ via concatenation, and finally select $2n$ to the training rollout set. Both the augmentation and the selection process introduce no additional computational cost. Therefore, the rollout generation time of Octopus is similar to that of the baseline with $n$. We set $n=8$ in the paper.
>
> - **We report total training time per step in the paper.**
>   As shown in Table 2, the rollout generation time of Octopus with $n=8$ is comparable to that of GRPO and GSPO with $n=8$, and much lower than the baselines with $n=16$. Relative to the $n=8$ baselines, the only extra cost in Octopus comes from policy update; relative to the $n=16$ baselines, Octopus saves more than half of the rollout generation time. As a result, the total training time per step of Octopus (958.1s) is close to the baselines with $n=8$ (GRPO: 845.1s, GSPO: 753.1s), while remaining substantially lower than the baselines with $n=16$ (GRPO: 1428.4s, GSPO: 1322.8s). Despite this, Octopus achieves sota performance under both rollout settings. For clarity, we also reproduce the results from Table 2 below, where **Total training time/step** includes both rollout generation and policy update.
>
>     |Methods|Rollout generation time(s)/Step|Total training time(s)/Step|Inference accuracy|
>     |-|-|-|-|
>     |GRPO(n=8)|364.9|845.1|68.0|
>     |GRPO(n=16)|679.1|1428.4|69.1|
>     |GSPO(n=8)|361.7|753.1|66.6|
>     |GSPO(n=16)|688.9|1322.8|70.7|
>     |SRPO(n=8)|410.9|895.8|67.4|
>     |SRPO(n=16)|816.6|1543.3|68.3|
>     |Ours|344.7|958.1|71.7|
>
> ---
>
> > W2 & Q1: Reward Hacking
>
> - **Reward hacking is not introduced by Octopus.**
>   In our setting, reward hacking comes from the conflict between the goals of direct reasoning and self-correction, rather than from our algorithm itself. Prior work [1][2] on self-correction also faces this issue and typically relies on reward shaping to solve it. But we show in the paper that reward shaping alone can lead to reward hacking in the middle and later stages of training.
>
> - **Reward hacking arises because the model exploits the easiest way to maximize reward.**
>   As training progresses, the model may discover shortcut behaviors that increase reward without improving the intended capability. In the self-correction setting, the model typically receives a relatively low accuracy reward in the early stage of training. At this stage, improving reward through better direct reasoning and genuine self-correction is comparatively easy. However, in the later stage of training, when the model seeks to further maximize reward, it may discover a shortcut: first intentionally producing an incorrect answer, and then generating the correct one, thereby obtaining a high reward score. This undesirable pattern can gradually accumulate during training and eventually lead to reward hacking.
>
>
> - **Octopus mitigates reward hacking by disentangling the training objectives.**
>   Octopus separates the learning signals for direct reasoning and self-correction through the proposed two-stage training framework and Response-Masking Strategy. This design prevents reward hacking and enables stable training, while continuously improving both direct reasoning and self-correction ability.
>
> ---
>
> > Q2: Why SFT?
> - We include an SFT cold-start stage to ensure the model follows the self-correction response format $o_1\oplus \texttt{\<sc\>}\oplus o_2$. A natural alternative is to simply prompt the model to respond in this format. However, as discussed in Section 4.1, the model often generates an $o_2$ that merely continues or partially revises $o_1$, leading to incomplete post-correction reasoning. Pairing such an $o_2$ with a different $o_1$ would produce incoherent augmented samples. Therefore, we introduce an SFT cold-start stage to ensure that both $o_1$ and $o_2$ are generated from scratch in a stable and reliable format. This stage is mainly for format learning rather than capability improvement, which is also supported by Table 3, where model performance changes very little after SFT.
>
> ---
>
> # Reference
> [1] Training Language Models to Self-Correct via Reinforcement Learning (ICLR 2025)
>
> [2] SRPO: Enhancing Multimodal LLM Reasoning via Reflection-Aware Reinforcement Learning (NeurIPS 2025)
>
> [3] VL-Rethinker: Incentivizing Self-Reflection of Vision-Language Models with Reinforcement Learning (NeurIPS 2025)
>
> [4] Sherlock: Self-Correcting Reasoning in Vision-Language Models (NeurIPS 2025)
>
> [5] Vision-Language Models Can Self-Improve Reasoning via Reflection (NAACL 2025)

---

> > ### Author Rebuttal · Reviewer_fUWh · 2026-04-07
> >
> > The authors provide comprehensive results to support the efficiency of their self-correction algorithm training compared to baseline RL methods. In addition, the authors also provide explanation for justifying why SFT is necessary in the training stage. Since I have already given a relatively positive score, I will keep my score as it is.

---

> > > ### Author Response · Authors · 2026-04-08
> > >
> > > Thank you very much for your positive feedback. Your insightful comments have greatly improved our work. We will revise our manuscript based on your suggestion. We sincerely appreciate your support!

---

### Official Review · Reviewer_GBi9 · 2026-03-12

**Soundness:** 3
**Presentation:** 3
**Significance:** 2
**Originality:** 2
**Overall Recommendation:** 4
**Confidence:** 4

**Summary:**

This work proposes Octopus, Correction-Specific Rollouts, a rollout augmentation framework that simultaneously improves sample efficiency and training stability via curating a format from responses before and after self-correction. It first identifies that the effective self-correction signal in the standard RL pipeline is sparse, whereas Octopus is a natural way to utilize the rollout generations to augment training samples. Putting together, it involves a two-stage RL training: in stage 1, it only updates the correction behavior; in stage 2, it masks tokens that receive conflicting reward signals in order to co-evolve both direct reasoning and self-correction behaviors. Experimental results show that Octopus-8B outperforms baselines in a wide selection of multimodal benchmarks. In addition, it provides a thorough ablation to components in Octopus and the training strategies.

**Compliance With Llm Reviewing Policy:**

Affirmed.

**Final Justification:**

Thanks to the author for their detailed response. It has addressed most of my concerns, and I will keep my rating towards acceptance.

**Key Questions For Authors:**

1. Is there any particular reason that we limit ourselves to the multimodal benchmarks? Does it also hold in the general text reasoning domains with text-only LLM backbones?
2. Since the thinking tokens can be very long for both o1 and o2, how to handle the cases when o1+o2 exceeds the context length?
3. What’s the training overhead for the number of additional rollouts? And how about the inference overhead?

**Limitations:**

It will be helpful to include a limitation section.

**Strengths And Weaknesses:**

Strength:

1. I am generally in favor of the idea to reuse the rollouts in constructing a self-correction trace that can explicitly guide the LLM to learn self-correction behaviors.

2. The method itself does not introduce computation overhead for the current batch.

3. The empirical results are extensive, and I find the improvement is solid. The ablation study and analysis are solid and fruitful to read.

Weaknesses:

1. I am unsure that this work focuses entirely on VLMs. Text-only reasoning should also be considered in this context. Otherwise, it raises concerns whether self-correction that leads to improvement with this rollout augmentation is simply a VLM-specific behavior in target visual domains.

2. While self-correction is a desirable behavior, I think this work currently misses a motivation for why we need a self-correction behavior, rather than having a longer output length for thinking.

3. The response length analysis is missing, and the inference latency is also unclear. I actually expect a plot of response vs. training steps. A reward vs steps plot against baseline plot is also missing.

4. This work studies under a relatively small output length. Hence, it is unclear whether the method will encounter situations where o1+o2 exceeds the context length.

---

> ### Author Rebuttal · Authors · 2026-03-31
>
> Thanks for your valuable feedback.
>
> ---
> > W1&Q1
> - **Our paper focuses on VLMs.**
> Our goal is to improve reasoning in VLMs through better self-correction. We focus on VLMs because their training and evaluation settings are relatively unified: training mainly uses ViRL-39K, while evaluation focuses on benchmarks in a VQA format with verifiable answers. Moreover, the most closely related baseline, SRPO [2], also targets VLM reasoning. It makes VLMs the most appropriate setting for a fair comparison.
> - **The learned abilities generalize to text-only tasks.**
>  Although we do not study text-only RL in the paper, Octopus naturally generalizes beyond multimodal settings. The table below shows that Octopus-8B outperforms the baseline on text-only benchmarks (AIME2024, MMLU-Pro), despite being trained only on image-text data. This suggests that the direct reasoning and self-correction abilities learned by Octopus are not limited to VLMs, and may also benefit text-only RL.
> |Methods|MMLU-Pro|AIME2024|
> |-|-|-|
> |Qwen3-8B-VL-Instruct|71.6|40.0|
> |GRPO|80.0|33.3|
> |GSPO|81.4|30.0|
> |SRPO|80.0|33.3|
> |Ours($o_1$)|81.4|40.0|
> |Ours($o_2$)|85.7|46.7|
> ---
> > W2
>
> Self-correction can be viewed as extending the thinking process with correction tokens, and our results show that it is more effective than simply generating longer outputs.
> - **Standard reasoning VLMs produce self-corrective tokens, but not effective correction.**
> Standard reasoning VLMs often produce tokens such as “wait” during reasoning. A natural idea is to append such tokens at inference time to encourage longer reasoning. However, as shown in Table 1, this does not improve performance. We believe this is because self-correction is neither explicitly supervised nor rewarded during training, so the model doesn't learn to use these tokens for effective correction.
> - **Self-correction enables reliable test-time scaling (tts).**
> Figure 7 shows that stronger self-correction leads to reliable tts. For Octopus-8B, appending more <sc> tokens produces longer responses and improves accuracy. In contrast, under similar inference token budgets, simply generating more tokens with standard RL baselines doesn't bring further gains, as shown in the table below. This suggests that effective self-correction, rather than longer output alone, is the key to tts.
>     |Methods|Token Number|Inference Time|Acc|
>     |-|-|-|-|
>     |GRPO+"wait"|1119|173|72.9|
>     |GSPO+"wait"|1130|177|72.7|
>     |SRPO+"<reflection>"|1091|171|73.4|
>     |Ours|1063|166|75.2|
> ---
> > W3&Q3
> - **Training and inference latency.**
> As shown in the table below, although Octopus includes an explicit self-correction stage, it adds only modest inference overhead. On MathVista, it achieves higher accuracy with inference time comparable to the baselines.
>     |Methods|Rollout generation time(s)/Step|Total training time(s)/Step|Inference time(s)|Inference acc|
>     |-|-|-|-|-|
>     |GRPO(n=8)|364.9|845.1|163|79.1|
>     |GRPO(n=16)|679.1|1428.4|160|80.8|
>     |GSPO(n=8)|361.7|753.1|167|78.5|
>     |GSPO(n=16)|688.9|1322.8|156|81.0|
>     |SRPO(n=8)|410.9|895.8|133|77.9|
>     |SRPO(n=16)|816.6|1543.3|148|79.8|
>     |Ours|344.7|958.1|166|82.1|
> - **Training curves.**
> We report the training curves of response length and validation accuracy reward here. Octopus generates longer responses early in training, but the response length gradually converges and becomes comparable to GSPO and GRPO after about 100 steps. Meanwhile, its validation accuracy reward improves steadily and eventually surpasses the baselines.
> |Response Length|1|81|161|241|321|401|
> |-|-|-|-|-|-|-|
> |GRPO|2291|1913|1885|1784|1717|1674|
> |GSPO|2283|1880|1750|1538|1500|1452|
> |SRPO|2299|1428|1500|1331|1279|1207|
> |Ours|2502|1692|1787|1865|1684|1697|
> |Acc Reward MMK12|1|81|161|241|321|401|
> |GRPO|65.6|69.6|73.7|72.7|75.1|76.3|
> |GSPO|65.6|72.1|75.1|76.7|78.0|78.8|
> |SRPO|61.5|64.9|70.0|73.9|73.1|75.4|
> |Ours|64.3|73.8|76.4|77.3|78.3|79.8|
> ---
> > W4&Q2
> - **Our response-length setting is reasonable**
> We set the response length to 6144 following prior work on ViRL-39K [3]. With Qwen3-VL-8B-Instruct as the base VLM, the truncation rate is only about 10%, and most truncated cases are due to repetitive generation rather than insufficient length, suggesting that this setting is suitable for our task.
> - **Truncation is a general issue in RL**
> In RLVR, overlong truncated responses are treated as negative samples and thus naturally discouraged during training. In our setting, even after concatenation, the initial truncation ratio remains only around 10%. If truncation happens due to repetition, this should be viewed as a general RL training issue rather than a problem specific to Octopus.
> - **Longer responses may further help**
> Our current RL is run on one node with 8 H100 GPUs, which limits the response length. With more resources, longer responses and stronger thinking models may further improve performance, which we leave for future work.
> ---
> # Reference
> Please see response to Reviewer fUWh.

---

> > ### Author Rebuttal · Reviewer_GBi9 · 2026-04-03
> >
> > Thanks to the author for their detailed response. It has addressed most of my concerns, and I will keep my current rating.

---

> > > ### Author Response · Authors · 2026-04-04
> > >
> > > Thank you for your constructive suggestions and your acknowledgement. We sincerely appreciate that our rebuttal has helped address all of your concerns. If there are any remaining questions or suggestions for improvement, we would be grateful to receive them.

---

### Official Review · Reviewer_PwLn · 2026-03-13

**Soundness:** 3
**Presentation:** 3
**Significance:** 2
**Originality:** 1
**Overall Recommendation:** 4
**Confidence:** 3

**Summary:**

This paper introduces Octopus, a method designed to train the self-correction capabilities of Vision-Language Models (VLMs) by effectively restructuring data for reinforcement learning. Octopus artificially generates various data combinations and extracts the desired pairs to construct a training set. To address challenges encountered during the subsequent learning process, the authors employ several strategies: Cold-Start and Data Construction, Mixed Sampling, Reward Shaping, and a Response-Masking Strategy. The experiments evaluate the model on math-related problems and general tasks. The results demonstrate that, except for the Math Verse task, the proposed method achieves superior performance over baselines without incurring additional costs.

**Compliance With Llm Reviewing Policy:**

Affirmed.

**Final Justification:**

The paper is well-written and follows a clear logical progression, particularly in how it introduces the proposed Octopus methodology and addresses subsequent engineering challenges. The authors identify data scarcity as a critical bottleneck, and their attempts to mitigate this issue are noteworthy.

While the experiments are well-designed, the proposed method lacks a strong theoretical foundation and exhibits limited novelty. Although the rebuttal successfully clarified that the method does not increase inference time—correcting a previous misunderstanding—the overall performance gains remain marginal. Furthermore, the submission lacks a thorough justification for the necessity of GSPO over existing alternatives. Consequently, I have decided to maintain my original score.

**Key Questions For Authors:**

- The performance of this model seems highly dependent on the quality and combination of rollouts. If the rollout quality is low or the combinations are suboptimal, the performance likely degrades significantly. Has any additional experimentation been conducted to analyze the sensitivity of the model to rollout quality?

**Limitations:**

yes

**Strengths And Weaknesses:**

### Strengths
- The methodologies proposed to address issues in the self-correction process (Cold-Start, Mixed Sampling, Reward Shaping, and Gradient Masking) appear technically sound.

- The experiments are well-designed, providing a clear and effective comparison with baseline models.

- The paper correctly identifies data scarcity as a core challenge in self-correction reinforcement learning.

- The manuscript is well-structured and easy to follow, logically presenting specific problems associated with certain methods and their respective solutions.

### Weaknesses
- There are lingering concerns regarding the actual efficiency of this method. Since an additional inference step consumes more time and computational resources, the performance gains compared to single-inference baselines do not seem particularly substantial. It could be argued that focusing on training the model to produce a correct inference in a single attempt might be a more effective approach.

- Further clarification is needed on why GSPO was selected as the underlying algorithm for Octopus. It remains unclear whether this choice was made simply because GSPO is generally or empirically effective, or if it possesses specific characteristics uniquely suited for this task.

- While the paper suggests a data recombination method to solve the data scarcity problem, its originality appears somewhat limited compared to existing approaches.

---

> ### Author Rebuttal · Authors · 2026-03-31
>
> Thank you for your great efforts on the review and constructive comments.
>
> ---
> > W1: Inference efficiency
>
> - **Octopus does not introduce additional inference steps.**
>   Octopus performs inference in a single pass, as in standard VLMs. The key difference is that Octopus explicitly formalizes the self-correction trigger as a fixed special token. Octopus directly generates an output of the form $o_1 \oplus \texttt{\<sc\>} \oplus o_2$, where \<sc\> acts as an explicit self-correction token, and $o_1$ and $o_2$ are the responses before and after correction. Since both $o_1$ and $o_2$ are generated from scratch within the same pass, Octopus requires only a single inference step.
>
> |Methods|Inference time(s)|Inference accuracy|
> |-|-|-|
> |GRPO(n=8)|163|79.1|
> |GRPO(n=16)|160|80.8|
> |GSPO(n=8)|167|78.5|
> |GSPO(n=16)|156|81.0|
> |SRPO(n=8)|133|77.9|
> |SRPO(n=16)|148|79.8|
> |Ours|166|82.1|
>
> - **Octopus improves accuracy with similar inference efficiency.**
> As Octopus generates an output of the form $o_1 \oplus \texttt{\<sc\>} \oplus o_2$, one might wonder if this incurs a higher inference cost. As shown in the table above, Octopus does not introduce significant inference overhead. On MathVista, its inference time is only slightly higher than that of baselines such as GSPO, mainly because it generates slightly more tokens on average. Despite this small increase in latency, Octopus achieves clearly higher accuracy. These results show that Octopus improves model performance while preserving strong inference efficiency.
>
> ---
> > W2: Why GSPO?
>
> We chose GSPO because it stabilizes Octopus training. GSPO employs a sequence-level importance weight, $s_i(\theta) = \frac{\pi_\theta(o_i \mid x)}{\pi_{\text{old}}(o_i \mid x)}$, which mitigates training collapse caused by a small number of excessively off-policy tokens in long reasoning trajectories and thus stabilizes RL optimization. In our framework, although both $o_1$ and $o_2$ are generated on-policy by the actor model, the augmentation inevitably introduces some degree of off-policy effect. GSPO is therefore a more suitable choice, as it provides greater training stability in this setting. We will revise our paper in Preliminaries section to clear this point.
>
> ---
>
> > W3: Lack of Originality
> - **Novelty of Octopus augmentation.**
>   The key insight of Octopus is that generated rollouts already contain useful self-correction signals, which can be constructed from pairs of incorrect and correct responses. Based on this observation, Octopus fully exploits the information within existing rollouts to obtain richer training signals, without introducing any additional computational cost. To our knowledge, no prior work has identified or exploited this structure within RL rollouts.
>
> - **Simple yet effective.**
>   We view the simplicity of Octopus as a strength, not a limitation. Octopus is built on a simple idea: rollout augmentation can be achieved by merely concatenating responses, without any complex extra computation. Despite this lightweight design, it effectively improves the model’s reasoning ability and achieves state-of-the-art performance across seven benchmarks while using only $0.72\times$ the training time of the baselines.
>
> We would appreciate if the reviewer could specify the existing approaches they consider similar, so that we can provide a more detailed comparison.
>
> ---
>
> > Q1: Rollout combination sensitivity
> - **Performance is not sensitive to rollout quality or combination.**
> As described in Section 3.2 and Algorithm 1, we do not select augmented rollouts based on reasoning quality. Instead, we only control the ratio of positive/negative samples in each batch and the proportion of samples containing valid self-correction signals. This selection is largely random and does not intentionally favor higher-quality $o_1$ or $o_2$. Yet this simple strategy already achieves superior performance over prior baselines.
>
> - **Even fully random augmentation improves performance.**
>   In the ablation study (Table 3), we compare three settings: no augmentation, fully random selection of augmented samples, and the selection criterion used in Octopus. Even with fully random selection, the model still achieves substantially better average performance across seven benchmarks (68.6) than training without augmentation (67.4). Moreover, the selection criterion in Octopus introduces no additional GPU computation overhead: it only performs lightweight random selection based on the positive/negative ratio and the proportion of valid self-correction samples, yet already achieves state-of-the-art performance.
>
> We agree that the current combination strategy may not be optimal, and developing better strategies for selecting higher-quality augmented samples is an interesting direction for future work.

---

> > ### Author Rebuttal · Reviewer_PwLn · 2026-04-04
> >
> > W1: Upon further review of the "Time" column in Table 2, I have clarified my previous misunderstanding. I now recognize that the proposed method enhances performance without incurring additional temporal costs during the inference phase, as clarified by the authors. I acknowledge the efficiency of the framework in this regard.
> >
> > W2: While I appreciate the additional explanation regarding GSPO, the current justification remains insufficient. The argument that GSPO was selected simply because $o_1$ and $o_2$ are generated on-policy while data augmentation induces off-policy effects feels somewhat perfunctory. A more rigorous theoretical or comparative analysis is needed to explain why GSPO specifically—rather than other hybrid or off-policy algorithms—is uniquely suited for this self-correction task.
> >
> > W3: Although the proposed approach represents a novel attempt that has not been explicitly presented before, the degree of architectural or conceptual innovation appears incremental rather than transformative. While the method is effective, the marginal gain over strong baselines raises questions about its long-term scalability across more complex multimodal tasks.
> >
> > Q1: I have noted the authors' explanation regarding the impact of rollout quality and combinations. Thank you for the clarification.

---

> > > ### Author Response · Authors · 2026-04-05
> > >
> > > Thank you for your continued engagement and thoughtful feedback. We sincerely appreciate that our rebuttal has helped address all of your concerns.
> > >
> > > ---
> > >
> > > > W2
> > >
> > > In our study, we also experimented with GRPO and DAPO as the RL algorithms. Both methods calculate importance-weight and conduct clipping at the token level. We found that when the amount of off-policy signal in the training batch increases, token-level clipping leads to training instability: in the later stage of training, the model experiences training collapse, with both entropy and gradient norm exploding. In contrast, GSPO performs clipping at the sequence level, directly filtering overly off-policy sequences rather than individual tokens. Empirically, we find that after augmentation, the overall clipped token ratio (~20%) under sequence-level importance weighting remains similar to that of the original rollout set before augmentation. This suggests that GSPO does not over-clip and is better aligned with the Octopus framework. We will add corresponding explanations and supporting results in the revision.
> > >
> > > ---
> > >
> > > > W3
> > >
> > > We would like to clarify that the improvement of Octopus-8B is substantial rather than incremental:
> > >
> > > - It outperforms the base model by 9.5-point on average.
> > > - Under a comparable training-time budget (rollout $n=8$), it outperforms RL baselines by 3–5 points on average.
> > > - It even outperforms the strongest baseline with rollout $n=16$ by 1.0 point while requiring only 0.72x the training cost.
> > > - It enables test-time scaling via additional $\texttt{\<sc\>}$ tokens, where performance continues to improve.
> > > - To the best of our knowledge, it achieves state-of-the-art performance among open-source VLMs across all 7 benchmarks.
> > >
> > > We believe Octopus could be further improved under a training budget comparable to baselines with rollout $n= 16$, for example by selecting more augmented samples or increasing the number of original rollouts. We leave this as future work.
> > >
> > > Finally, we note that the contribution of this paper is a new RL training paradigm for self-correction. The insight that self-correction signals can be explicitly constructed from existing rollouts without additional compute is conceptually simple and practically impactful.

---

### Decision · Program_Chairs · 2026-04-30

**Decision:**

Accept (regular)

**Comment:**

This paper proposes Octopus, a framework to enhance self-corrective reasoning in VLMs without adding computational overhead. During the rebuttal period, the authors actively engaged with reviewers, successfully resolving misunderstandings regarding inference efficiency, demonstrating text-only generalization, and clarifying how their method mitigates reward hacking and maintains on-policy generation. Reviewers highlighted the zero-cost data augmentation, high computational efficiency, and robust performance as key strengths, while noting drawbacks such as incremental conceptual novelty, weak theoretical justification for the use of GSPO, and the risk of logically mismatched reflection points when randomly pairing incorrect and correct reasoning trajectories.

Ultimately, the AC believes the paper's practical merits and strong empirical outcomes outweighh its flaws and recommend acceptance. The concerns regarding incremental novelty are subjective, and while deeper theoretical grounding for the algorithm choice is nice to have, this is ultimately an effective empirical paper. However, some reviewer criticisms remain highly valid—particularly the logical inconsistencies that can arise from randomly pairing disparate trajectories—and the authors must ensure they carefully address these limitations and remaining feedback in the final camera-ready version.